# A joint diffusion/collision model for crystal growth in pure liquid metals

Hua Men ⬤[1] ✉

The kinetics of atomic attachments at the liquid/solid interface is one of the foundations of solidification theory, and to date one of the long-standing questions remains: whether or not the growth is thermal activated in pure liquid metals. Using molecular dynamics simulations and machine learning, I have demonstrated that a considerable fraction of liquid atoms at the interfaces of Al(111), (110) and (100) needs thermal activation for growth to take place while the others attach to the crystal without an energy barrier. My joint diffusion/collision model is proved to be robust in predicting the general growth behaviour of pure metals. Here, I show this model is able to quantitatively describe the temperature dependence of growth kinetics and to properly interpret some important experimental observations, and it significantly advances our understanding of solidification theory and also is useful for modelling solidification, phase change materials and lithium dendrite growth in lithium-ion battery.

It is generally believed that there is no thermal activation barrier for crystal growth in pure liquid metals[1,2]. Following the unsuccessful attempts to produce amorphous pure metals, Turnbull[3] proposed a collision-limited theory to explain this phenomenon: liquid atoms attached to the crystal with a ballistic velocity (at the speed of sound in the liquid) at the interface in the absence of a thermal activation barrier in the growth. Jackson[1] adopted an average thermal velocity of liquid atoms in the interface kinetic coefficient for the collision growth model instead, which could fit the molecular dynamics (MD) growth rates of a Lennard-Jones system with a (100) interface[4]. The revised collision-limited theory is supported by various experiments[5,6], for example the measured crystallization velocities can reach 100 m s⁻¹ and is too large to be the result of a diffusion-controlled crystallization process for pure metals (FCC Au, Cu[5], Al, Co[6], BCC Ti, Fe and HCP Zr[6]). The crystallization of pure Pb at a temperature as low as 4 K is further evidence that supports the collision-limited model[7], where the atomic diffusion rate approaches zero. This should hold true for simple materials, such as metals, inert gases and some simple molecular materials, since they do not have any rotational entropy in the liquid, and the crystallization of an atom does not depend on the structural rearrangement of other atoms at the interface in the liquid, so there is no thermal activation energy barrier for crystallization[1]. However, the collision model has been questioned even in the applications for pure liquid metals[4,8–12]. Another challenge to the collision theory was posed by a study of Zhong et al.[13]. With a high liquid-quench technique to achieve an ultrahigh cooling rate of 10¹⁴ K s⁻¹, they successfully produced nanometer-sized monatomic metallic glasses (MGs) of BCC metals (Ta, V, Mo), which is stable at ambient temperature, but not from pure FCC metals (Au, Ag, Cu, Pd, Al, Rh, Ir). The odd experimental observation was attributed to the difference in the growth kinetics, with crystallization of pure BCC metals as diffusion-controlled[13–16] and that of FCC metals as an almost barrier-less process[15] or with little or no activation energy[14].

A fundamental problem associated with this question is that the growth velocity, $V$, cannot be predicted properly with all the proposed models[4,17–21], such as the collision-limited[4], transition rate (diffusion-controlled)[19,20] and kinetic density function theories[21]. The growth velocity is governed by a driving force term, provided by the free energy difference between the liquid and solid, $\Delta\mu(T)$, and a kinetic term, $k(T)$, for atom attachment at the liquid/solid (L/S) interface[18]:

$$V = k(T)\left(1 - \exp\left(-\frac{\Delta\mu}{k_B T}\right)\right),\tag{1}$$

where $k_B$ is Boltzmann's constant and $T$ is absolute temperature. $\Delta\mu(T)$ increases almost linearly with an increase in undercooling, $\Delta T$, below

[1]BCAST, Brunel University London, Uxbridge, Middlesex UB8 3PH, UK. ✉e-mail: hua.men@brunel.ac.uk

the melting point. On the other hand, $k(T)$ decreases with increasing $\Delta T$, however its dependence on $\Delta T$ is highly relevant to the mechanism of atomic attachments at the interface. It is evident that the liquid atoms adjacent to the crystal can have a local intrinsic structure[22], which templates the lattice of the crystal surface but is hidden by thermal motion[18]. The attachments of such liquid atoms to the crystal will advance the interface a distance of $a$ without the thermal activation barrier, according to collision-limited theory[4]:

$$V = f\frac{a}{\lambda}\sqrt{\frac{3k_BT}{m}}\left(1 - \exp\left(-\frac{\Delta\mu}{k_BT}\right)\right), \qquad (2)$$

where $f$ is the density of growth sites, $\lambda$ the atomic displacement, and $m$ the atom mass. There exists a temperature dependence of the crystal growth velocity for pure metals, and atomistic simulations reveal that the maximum of the growth velocity is usually located at about $0.7T_m$, which is so-called crossover or turnover temperature ($T_c$)[18,23–25]. The velocity at small undercooling is often used to fit with Eq. (2) for justifying the collision theory[18,23,24]. However, the driving force dominates in the growth kinetics at small undercooling. Equation (2) fails to fit the simulated crystal growth data at larger undercooling due to its weak temperature dependence while more realistic potential is employed in the simulations[25]. Consequently, the collision growth mode can not properly predict the cross-over of the growth velocity for pure metals.

Being lack of a physical origin for pure metals though[1], the diffusion-controlled model is often used to fit the crystal growth data, as described by Wilson-Frenkel (WF) expression[1,19,20]:

$$V = fa\nu \exp\left(-\frac{Q}{k_BT}\right)\left(1 - \exp\left(-\frac{\Delta\mu}{k_BT}\right)\right), \qquad (3)$$

where $\nu$ is a frequency of order of the Debye frequency, and $Q$ is the activation energy for diffusion in the liquid. With a strong temperature dependency, Eq. (3) can fit the crystal growth data of some FCC and BCC metals with $C$ ($= fa\nu$) and $Q$ as the fitting parameters[25]. However, when the fitting parameters are replaced by measured values, the calculated velocity with Eq. (3) is too low to validate the WF model[18,26]. As a modification of the WF model, the local-structure dependent crystal growth model[27] indicates that significant structural ordering in the interfacial layers adjacent to the crystal effectively reduces the mobility of liquid atoms and slows down the crystallization kinetics. In the opposite, another recent study[26], where the WF model is modified by incorporating the effect of the liquid structural ordering, suggests that suppressing structural order at the L/S interface can reduce the growth velocity by several orders of magnitude. Additionally, the WF model seems to be supported by the experiment of successfully manufacturing the nanometre-sized monatomic BCC MGs[13–16], but fails to explain the fast growth of pure metals (including FCC and BCC) at $T_c$[5,6] and the crystallization of pure Pb metal at a very low temperature[7]. All these studies suggest that we still do not have a clear picture of the mechanism of atomic attachments in the crystal growth of pure liquid metals.

It is known that the crystal growth of pure metals proceeds by a structural templating mechanism (Supplementary 1. Structural templating mechanism), which works for heterogeneous nucleation as well[28,29]. The liquid atoms with pronounced atomic ordering at the L/S interface share common local potential energy minima with the surface of the crystal[18,30], and will template the lattice of the crystal in the growth. At the (111) interface of FCC metals, liquid atoms can template B or C positions on an underlying crystal atomic layer (assuming at A positions) (Supplementary Fig. 1). However, for growth to take place, atoms at C positions have to move to B positions for the normal ABCABC templating to be maintained consistently, leading to an energy barrier to overcome. On the other hand, atoms at B positions

can settle directly into the solid without any energy barrier. In other words, liquid atoms adjacent to a B site can attach to the solid by a direct collision process, but liquid atoms adjacent to a C site need to make a diffusional jump to a B site before or after attachment. Thus, thermal activation is needed for some liquid atoms to maintain consistent growth at least for the (111) interface, implying that collision and diffusion modes might both be involved in crystal growth of pure liquid metals. In this study, Al was chosen as the model system in that Al has a close-packed cubic structure as FCC metal, where the (111) plane is densely packed and the (110) and (100) planes are less densely packed. Thus, the anisotropy in the growth kinetics is only attributed to the structural factor, with the chemical effect excluded. The attachment mechanism of liquid atoms at Al (111), (110) and (100) interfaces is to be examined, using MD simulations, local bond-order analysis[31] and machine learning.

In this work, I establish the general growth kinetics for simple materials that a considerable fraction of liquid atoms at the interface is thermal activated in the growth, and develop an analytical joint diffusion/collision model, which can reasonably predict the crystal growth rate.

## Results

### Rough liquid/solid interfaces

The L/S interface of a pure metal is usually rough at an atomic-level, due to the contribution of configurational entropy[32]. The melting point, $T_m$, of Al(111), (110) and (100) interfaces has been determined to be 942 K, 940 K and 939 K, respectively, which is dependent of the anisotropy of crystallographic orientation due to the variation in the density of packing of atoms in a particular crystal plane. According to Lindemann's criterion[33,34], the Al(111) interface has the highest $T_m$. The Al (111) interface extends to about 6 atomic layers in term of atomic density profile, $\rho(z)$, at $T_m$. It is noted that the number of atoms per layer, $N_L$, increases continuously from the liquid to the solid across the interface (Fig. 1a). This is also true for Al(110) and (100) interfaces (Fig. 1b, c), as well as during crystal growth (Supplementary Fig. 2), in general agreement with the literature[8] (Supplementary 2. Densification at the interface). This phenomenon is consistent with the fact that the density of bulk liquid is usually a few percent lower than that of bulk solid. It indicates that some liquid atoms need to diffuse from bulk liquid to bulk solid across the L/S interface during crystal growth to maintain the mass conservation.

Figure 2 exhibits the morphology of an atomic-level rough (111) interface equilibrated at 942 K with Potential A (see Methods), where solid and liquid atoms are distinguished by employing machine learning with support vector machine (SVM) classifier[35–37] (Fig. 3 and Supplementary 3. Solid clusters in undercooled liquids). The L/S interface spans about 3, 5 and 3 atomic layers, respectively, at Al(111) (Fig. 2a), (110) (Supplementary Fig. 3a) and (100) interfaces (Supplementary Fig. 3b) in terms of the presence of solid atoms (Supplementary 4. Rough liquid/solid interfaces). The liquid atoms of the 1st (L1) to 3rd (L3) interfacial layers, sitting right above the underlying solid atoms (assumed to be at A positions), are generally situated either at B positions or on top of the Peierls barriers between solid atoms. However, those in the 4th (L4) interfacial layer can take up either B (enclosed by up triangle) or C positions (enclosed by down triangle) (Fig. 2b). This indicates that at equilibrium most of the liquid atoms at the interface has a local structure in registry with the solid (at B positions), with a fraction not in registry with the solid (at C positions, i.e., wrong templating).

Machine learning was used to characterize the behaviour of the atoms at the L/S interface. A local structure fingerprint, $\mathbf{x}_i$, for each atom $i$ is constructed from a set of 23 radial structure functions, $G_i(r)$[38,39], (See Methods). To create a two-dimensional (2D) representation of the $R^{23}$-space of $\mathbf{x}_i$, I performed linear dimensionality reduction with principal component analysis (PCA)[40]. As shown in

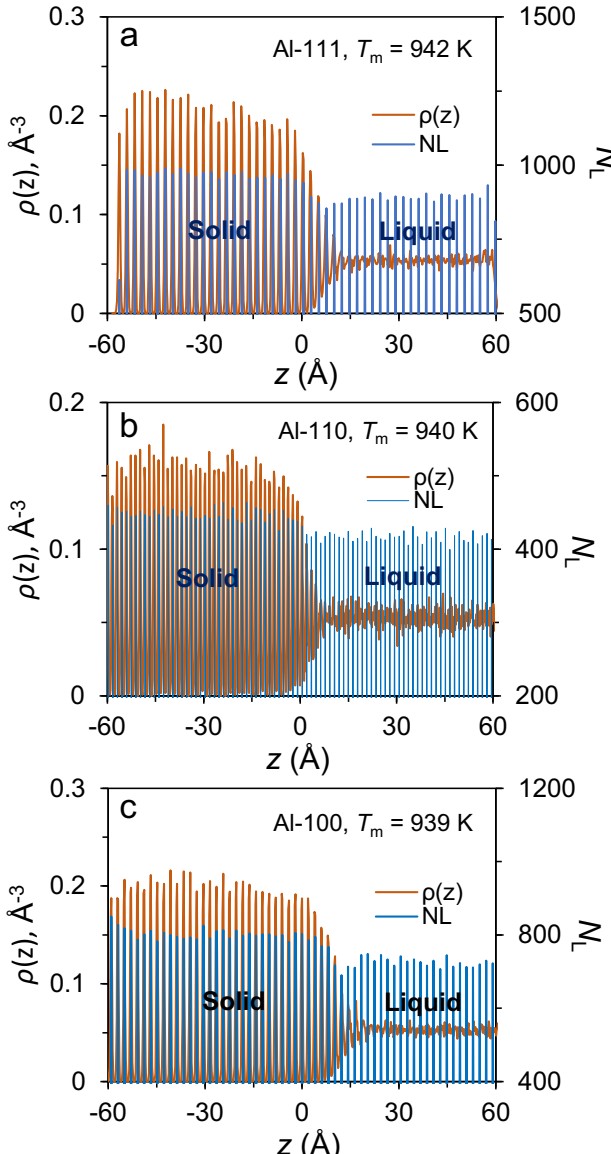

**Fig. 1 | Variation in number of atoms per layer at the interface at equilibrium.** Density profile, $\rho(z)$, and number of atoms per atomic layer, $N_L$, are plotted as a function of distance $z$ in the liquid/solid (L/S) Al systems with (**a**) (111), (**b**) (110) and (**c**) (100) interfaces, equilibrated at melting point ($T_m$) with Potential A (embedded-atom method (EAM) potential for aluminium, developed by Zope and Mishin (see Methods)). The $N_L$ gradually increases from bulk liquid to bulk solid across the interface at equilibrium. (Source data are provided as a Source Data file).

Fig. 4a, the solid and liquid regions are slightly overlapped in the PCA projection along first and second principal components for the Al(111) system equilibrated at 942 K, where solid atoms in the bulk phase/interface (denoted as solid), liquid atoms in the bulk phase (denoted as liquid) and liquid atoms at the interface (denoted as interface) are labelled according to an order parameter of $\alpha$ calculated with the local bond-order analysis[31] (Supplementary 5. Local bond-order analysis). Further, I carried out nonlinear dimensionality reduction with $t$-distributed Stochastic Neighbour Embedding ($t$-SNE) method[41,42], which is particularly sensitive to the local structure of the data. The liquid and solid regions are well separated in the $t$-SNE plots at equilibrium (Fig. 4b) or in the growth (Fig. 4c, d). It is interesting to note that the interface liquid atoms are located in either solid or liquid regions of the $t$-SNE plots. These atoms have the structures analogous to solid or liquid, and none of them is partially solid and partially liquid at either

the static rough interface or during growth, agreed with Jackson's suggestion[1].

### Crystal growth at Al(111) interface

The crystal growth velocity $V$ was calculated by two methods[8,25], described in detail in Methods. Figure 5a shows $V$ as a function of $\Delta T$ for the Al(111) system with Potential B (see Methods). $V$ increases initially with an increase in $\Delta T$, reaches a maximum at a crossover undercooling, $\Delta T_c$, of 230 K and then decreases. Here, $\Delta T_c \approx 0.25 T_m$. At small undercooling ($\Delta T < \Delta T_c$), the crystal growth exhibits a normal templating sequence of FCC structure (ABCABCABC…) (Supplementary Fig. 4a), and stacking faults (ABCACABCA…) are observed while $\Delta T$ is just larger than $\Delta T_c$ (Supplementary Fig. 4b), and twin boundaries (ABCBACBA…) (Fig. 5b) form during crystal growth at higher undercooling ($\Delta T > \Delta T_c$). The formation of defects in the growth appears to be related to the growth kinetics at corresponding undercooling, which in turn determines the growth velocity.

### Thermal activation in crystal growth

To establish the growth kinetics, I firstly examined the evolution of atomic arrangements in the interfacial atomic layers of the Al(111) system with Potential B during growth at high undercooling ($\Delta T > \Delta T_c$). At $\Delta T = 250$ K, stacking fault starts to form at the L/S interface, attributed to the wrong templating (region II in L3/L2 at $t = 0.16$ ns in Fig. 6a), and dies out later as the region II is overgrown by normal templating (region I in Fig. 6b, c) and finally eliminated by dislocation motions (Fig. 6d). At $\Delta T = 300$ K, twin boundary starts to form, where the wrong templating (region II) overgrows the normal templating (region I) (Fig. 7). In both cases, the dislocation motion involves to partially or completely eliminate the stacking faults, and thermal activation is necessary to overcome the energy barrier of dislocation motions, referred to as the dislocation correction mechanism.

Further, I investigated the atomic displacements at the interface at small undercooling ($\Delta T < \Delta T_c$), where stacking faults and twin boundaries were not observed. Here, only the 1st interfacial layer (L1) was taken into account so as to rule out the self-diffusion in bulk liquid. Figure 8a shows the snapshot of the L1 in the Al (111) interface at $t = 0.06$ ns during the simulation with $\Delta T = 160$ K with Potential A, which has a mixed structure with ordered and disordered regions. The L1 becomes fully solid at 0.069 ns, where the atoms are coloured according to the displacements, $d$, from 0.06 to 0.069 ns (Fig. 8b). There exist substantial atomic displacements in the L1 during the growth (Fig. 8c–e). About 3% atoms experiences the long-range diffusion with $d > d^{111}$, where $d^{111}$ is the atomic layer spacing of (111) orientation, and needs to overcome the energy barrier between two consecutive atomic planes. About 5% atoms has a displacement from C to B positions with $d^{CB} < d < d^{111}$, where $d^{CB}$ is the distance between C and B positions on (111) plane, and needs to jump from one local energy minimum to another in order to maintain the normal templating sequence in the growth (Fig. 9). In both cases, the atomic displacements are thermal activated, which is referred to as the displacement correction mechanism (Supplementary 6. Displacement and dislocation correction mechanisms). About 16% atoms has a displacement of $0.5 d^{CB} < d < d^{CB}$, where $d^{CB}$ is approximated to $0.5 d^{111}$, suggesting that these atoms have a local structure similar to the crystal but still in liquid status at 0.06 ns. They can escape from the local energy minimum and become solid without thermal activation barrier at 0.069 ns. The remaining atoms (about 76%) already become solid at 0.06 ns, vibrating at the equilibrium atomic positions with $d < 0.5 d^{CB}$.

Above analysis indicates that the considerable fraction of liquid atoms at the (111) interface needs thermal activation during the growth. About 13% of liquid atoms displaces across the Al(111) interface at $\Delta T = 160$ K, and another 21% needs to overcome an energy barrier between two local energy minima at the (111) plane. As a consequence, about one third of liquid atoms is thermal activated in the growth,

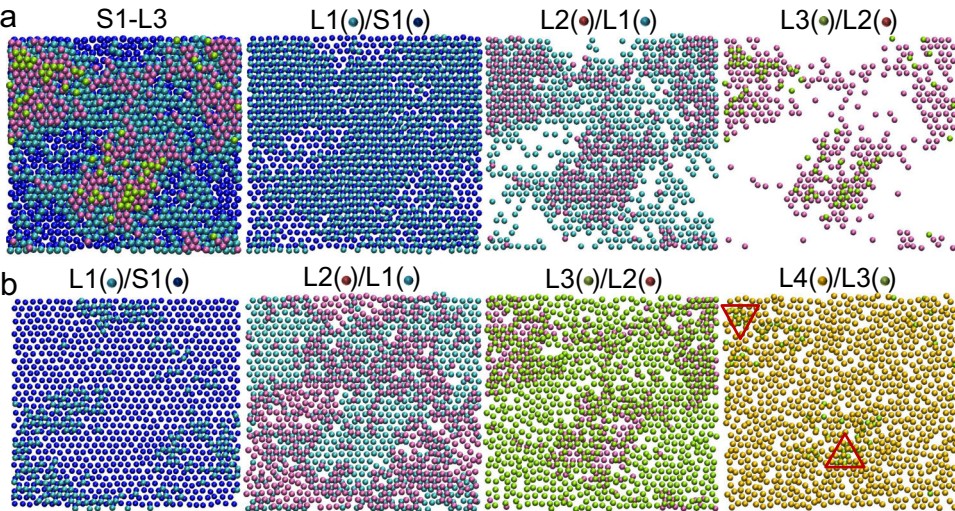

**Fig. 2 | Rough liquid/solid (111) interface. (a)** Solid and **(b)** liquid atoms in the 1st (L1) to 4th (L4) interfacial layers superimposed on solid atoms of underlying layer at the Al(111) interface equilibrated at $T = 942$ K with Potential A. S1 (blue spheres) denotes the 1st solid layer at the interface, and L1 (cyan spheres), L2 (mauve spheres), L3 (yellow spheres) and L4 (orange spheres) denote the 1st to 4th liquid layers, respectively. Solid and liquid atoms are identified using machine learning with support vector machine (SVM) classifier. The interfacial liquid atoms at the interface usually take a normal templating sequence of FCC(111) structure while sitting on solid atoms of underlying layer, and however the liquid atoms in the L4 can have either normal or wrong templating sequences (enclosed by up and down triangles, respectively, in **(b)**).

behaving in the diffusion-controlled mode. Meanwhile, the remaining 66% don't need thermal activation, following the collision-limited mode. Our results agree roughly with the literature, for instance it reports that about 20% of interfacial atoms undergoes a jump of less than an interatomic distance in growth[9], and 10% of atoms at (111) and (110) interfaces and 3% at (100) interface hop from a less dense layer to a denser layer at the interface in a LJ system during growth at $\Delta T = 60$ K[8].

**Predicting growth velocity**

To manifest the cooperation of collision-limited and diffusion-controlled growth models, I propose a joint diffusion/collision model to describe the growth kinetics:

$$V = f \frac{a}{\lambda} \sqrt{\frac{3k_B T}{m}} \exp\left(-\frac{Q}{k_B T}\right)^{x_{therm}} \left(1 - \exp\left(-\frac{\Delta\mu}{k_B T}\right)\right), \quad (4)$$

where $x_{therm} = b\Delta T + c$ is the fraction of liquid atoms with thermal activation in the growth, and $b$ and $c$ are constants. The details of model development can be found in Supplementary 7. Development of the joint diffusion/collision model. In general, the liquid atoms attach to the crystal with an average thermal velocity of $(3k_B T/m)^{1/2}$ in the case without thermal activation barrier, as Jackson suggested[1]. However, for these diffusion-controlled liquid atoms with a fraction of $x_{therm}$, the thermal activation will be required for the attachments at the interface with a probability of $\exp(-Q/k_B T)^{x_{therm}}$. The vibrational theory[18], as a modification of collision theory, suggests that the analogous vibrational frequency of the atoms in the crystal sets the timing for growth. I believe that initial status of the atoms before the attachments is relevant to the interface kinetics, rather than the status of the atoms after crystallization. Therefore, the average thermal velocity of liquid atoms is used in my model to feature the interface kinetics, which is manipulated by the fraction, $x_{therm}$, of the liquid atoms at the interface that have a thermal activation energy $Q$. In the general cases ($1 > x_{therm} > 0$), a fraction of $x_{therm}$ of liquid atoms is thermal activated and others ($1 - x_{therm}$) are athermal. My model reduces to the collision model for $x_{therm} = 0$ and the WF diffusion model for $x_{therm} = 1$. It should be pointed out that this model is applicable for all simple materials as no crystal structure-specific assumption is made during the development of the model.

With a fitting parameter of $x_{therm}$, the joint diffusion/collision model can reasonably predict the growth velocity for the Al (111), (110) and (100) interfaces, as shown in Figs. 5a, 10a, where the fittings with collision and diffusion models are also included for comparison (Supplementary 8. Fitting crystal growth data). In the fittings with diffusion and my models, the activation energy of diffusion $Q = 280 \pm 70$ meV, obtained from the experimental measurements in bulk liquid Al[43], was used. My model gives a reasonable agreement with the simulated data. The $b$ obtained from the fitting is 0.00075, 0.00052 and 0.0005, respectively, for Al(111), (110) and (100) interfaces. The maximum of the simulated growth velocity is about 59 m s⁻¹ at $\Delta T = 240$ K for the (100) interface. The fitting parameter, $x_{therm}$, increases with increasing $\Delta T$ for all three interfaces, and the order $x_{therm}^{111} > x_{therm}^{110} > x_{therm}^{100}$ holds at any corresponding $\Delta T$ (Fig. 10b). For instance, the $x_{therm}^{111}$ increases from 0.24 at $\Delta T = 10$ K to 0.7 at ambient temperature for the (111) interface and meanwhile the $x_{therm}^{100}$ increases from 0.17 to 0.5 for the (100) interface. The increase in the $x_{therm}$ may be mainly attributed to the increase in the size and quantity of solid clusters in the undercooled liquid with increasing $\Delta T$ (Supplementary 3. Solid clusters in undercooled liquids). The $x_{therm}$ from the fitting is in an excellent agreement with the atomic displacement analysis above, e.g., $x_{therm}^{111} = 0.36$ at $\Delta T = 160$ K, compared to the percentage (34%) of liquid atoms that is thermal activated at a (111) interface in the growth. It is interesting to note the growth kinetic coefficient, $\mu$, has an inverse order for FCC metals: $\mu^{111} < \mu^{110} < \mu^{100}$[44,45], implying that physical origin of the anisotropy in $\mu$ might be relevant to the variation in the $x_{therm}$ with orientations in the crystal growth of pure metals.

My model also can fit the simulated growth velocity of BCC metals, such as Ta, with the data recovered from the study of Zhong et al.[13], as shown in Fig. 10c. The activation energy of diffusion for liquid Ta is $Q = 310$ meV, taken from ref. 25. The $b$ obtained from the fitting is 0.0004 for Ta(110) and (100) interfaces. It is noted that the $b$ is very close for the metals, especially for the less densely packed interfaces of either FCC or BCC metals. The growth velocity reaches the maximum of about 69 m s⁻¹ at $T_c = 2350$ K, which is about $0.7T_m$. On the other hand, the prediction with diffusion model exhibits the large

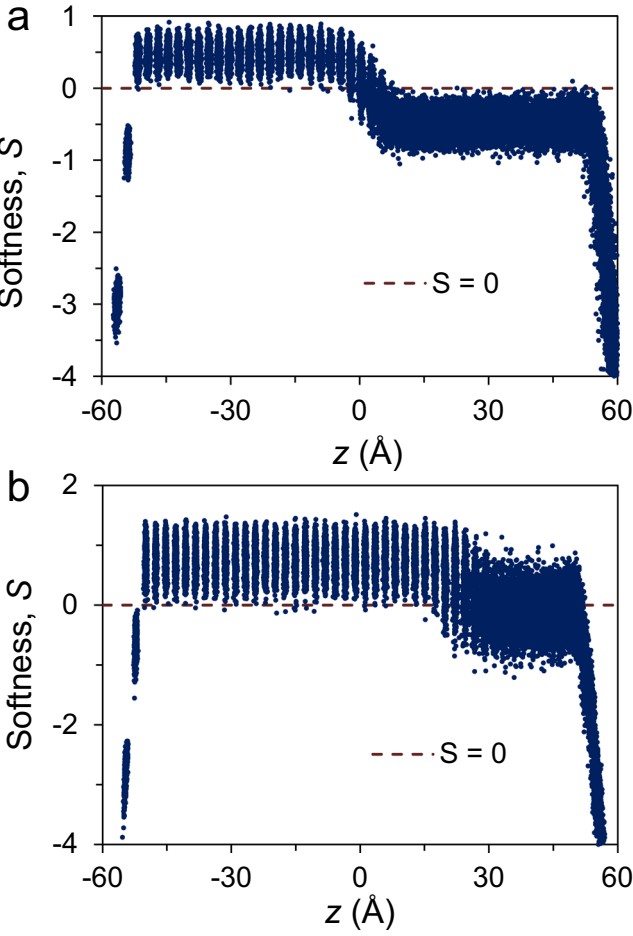

**Fig. 3 | Softness, $S$, calculated with support vector machine (SVM) model.** The $S$ is plotted as a function of the distance, $z$, for the Al(111) interface, (**a**) equilibrated at 942 K and (**b**) at the simulation time $t = 0.12$ ns during the simulation at 582 K with Potential A. The dashed line represents $S = 0$. The solid atoms in the bulk have $S > 0$ and those in two bottom atomic layers have $S < 0$ due to surface effect. The liquid atoms in the bulk have $S < 0$ at the melting point ($T_m$), but some have $S > 0$ during the crystal growth due to formation of the solid clusters in bulk liquid. (Source data are provided as a Source Data file).

divergence from the simulated growth velocity of Ta in a temperature range from $T_m = 3290$ K to about 1040 K. The fitting with my model suggests that the $x_{therm}$ increases from 0.1 at $T = 3280$ K ($\Delta T = 10$ K) with an increase in $\Delta T$, and reaches 1.0 at 1040 K (Fig. 10d), which is indicative of a transition of the growth kinetics from the joint diffusion/collision mode to the diffusion-controlled mode, denoted as at $T_{tran}$. The growth behaviour of Ta below $T_{tran} = 1040$ K can be described by the diffusion-controlled model (Fig. 10c). It should be pointed out that such a transition is solely attributed to high $T_m$ of Ta, which renders a larger $\Delta T$ to be reached before completion of the solidification, and the same transition should also be expected for other BCC and FCC metals with high $T_m$. In general, BCC and FCC metals exhibit very similar growth behaviours, in terms of the maximum growth velocity, relative crossover temperature and temperature dependence of growth velocity and so on. It demonstrates that the joint diffusion/ collision model is robust to describe the growth kinetics of pure metals.

## Discussion

In this study, I resolve a fundamental question in understanding the interface kinetics of crystal growth, namely the relative roles played by thermal activation and direct collision for attachment of liquid atoms

at the growing interface. It has been reported that thermal activation is needed in some specific cases, e.g., concerted displacements at an FCC(111) interface[4,10–12] and thermally activated jumps of a small fraction of liquid atoms at high undercooling with a defect annihilation mechanism[9,46]. Some studies[13–16] suggest that the crystal growth of BCC metals is diffusion-controlled while that of FCC metals follows collision-limited mode. In the present study, it has shown that many liquid atoms at the interface attach to the crystal without a thermal activation, in accordance with the collision model[1], but a considerable fraction of liquid atoms needs thermal activation to overcome an energy barrier in agreement with the diffusion-controlled model[19,20]. Thermal activation takes place for several reasons. Firstly, long-range diffusion is needed for some of the bulk liquid atoms to move to the interface as crystallization of each atomic layer takes place. Secondly, some atoms at the (111) interface have to make substantial atomic displacements to correct the wrong templating, ensuring the registry with the layer sequence at the growing interface. Both mechanisms should be working for the growth kinetics at high undercooling, where stacking fault and twin boundaries form due to the sluggish kinetics at large undercooling and the dislocation correction mechanism starts to work. In all cases, thermal activation is needed for growth to take place as a rearrangement of some local structures at the interface is required, with energy barriers approximately equal to that for bulk diffusion in the liquid[1]. It is noted that for the FCC structures only the first case is applicable for (110) and (100) interfaces since the wrong templating cannot happen at the interfaces except (111) interface, and consequently the $x_{therm}^{110}$ and $x_{therm}^{100}$ are smaller than the $x_{therm}^{111}$. For the BCC metals, all the low-index interfaces are less densely packed, similar to (110) and (100) interfaces of the FCC metals, and thereby the $x_{therm}$ of BCC metals is smaller than the $x_{therm}^{111}$ of FCC metals.

The joint diffusion/collision model genuinely reflects the underlying atomic growth mechanism, and is able to quantitively describe the temperature dependence of growth kinetics and the turnover of growth velocity for pure metals. This phenomenon cannot be interpreted with collision theory due to its weak temperature dependence of the kinetic term[25]. Sun et al.[18] proposes that it is attributed to the kinetic instability of the liquid with respect to crystallization, where unstable growth of local crystal order starts throughout the liquid at high undercooling. From the point view of diffusion model, Ashkenazy et al.[25] suggests the stress that develops at the L/S interface below $T_c$ dramatically reduces the activation energy of interface mobility, which essentially falls to zero for FCC metals, and literally the diffusion-controlled model turns to the collision-limited growth mode at high undercooling. In the present study, we found that some spikes of rough interface start free growth at high undercooling as soon as they reach the critical size of free growth at corresponding $\Delta T$[47]. At the L/S interface, the original planar growth changes to the multiple-spherical growth (Supplementary 4. Rough liquid/solid interfaces), leading to so-called liquid instability[18] or mechanical instability[25]. However, the transition from the planar growth to the multi-spherical growth only causes abnormally fast growth, which produces apparent slope discontinuity of the growth velocity[23–25], but does not slow down the interface kinetics. This mechanism may also result in an anode roughening process correlated with the beginning stage of lithium dendrite formation in Li-ion batteries[48], and our finding could be helpful to understand the growth of the lithium dendrites. It should be pointed out that the growth kinetics could be slowed down by homogeneous nucleation in the front of growing solid at high undercooling, and even the growth is blocked in a way similar to the columnar-to-equiaxed transition (CET)[49]. My recent study[50] reveals that homogeneous nucleation can occur in pure liquid Al at an undercooling of 457 K, which falls in the range of high undercooling (maximum 650 K used here). However, this phenomenon has never been observed during the crystal growth in our study or in literature[13], possibly due to the current simulation or experimental conditions that

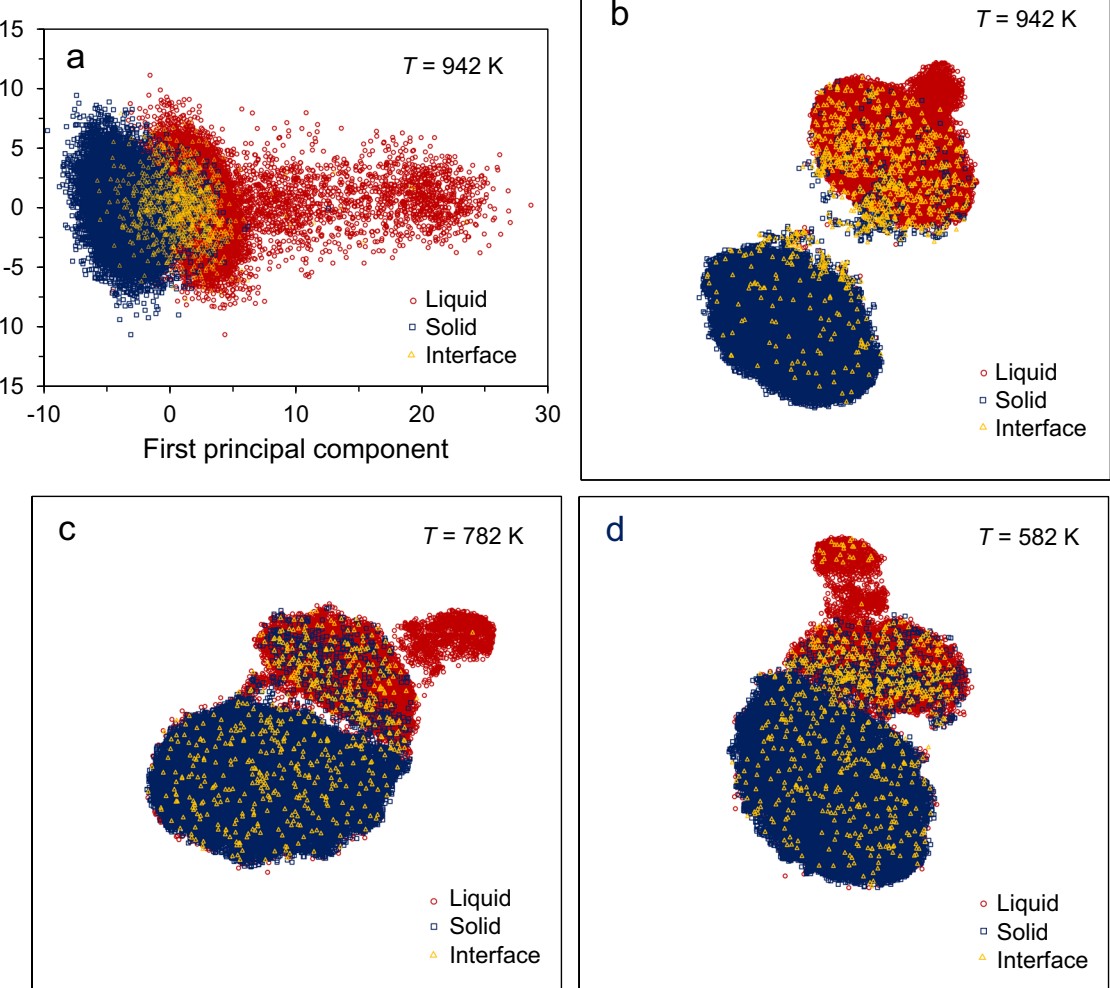

**Fig. 4 | Dimension reduction analysis.** (**a**) Two-dimensional principal component analysis (PCA) plot of $R^{23}$-space of the fingerprints, $\mathbf{x}_i$, in the Al (111) system equilibrated at 942 K with Potential A, and two-component $t$-distributed Stochastic Neighbour Embedding ($t$-SNE) plots (**b**) at equilibrium, (**c**) at the simulation time $t = 0.06$ ns with $T = 782$ K and (**d**) $t = 0.12$ ns with $T = 582$ K during the simulations. Solid, liquid and interface represent the solid atoms, liquid atoms in the bulk and liquid atoms at the interface, respectively, labelled by the order parameter $\alpha$ according to the local bond-order analysis. The solid and liquid atoms in the bulk are well separated in the $t$-SNE plots, and the liquid atoms at the interface can have the ordering of either bulk solid or liquid. (Source data are provided as a Source Data file).

the size of the simulation systems or experimental samples is limited to the nanometre scale. Some studies suggested that formation of the stacking fault involved a cooperative atomic movement in the growing layers[4,10–12], resulting in the slower kinetics at (111) interface than at (110) and (100) interfaces. However, the stacking fault mechanism alone may not play a major role in the growth kinetics[12], and works neither for FCC(110) and (100) interfaces (Supplementary Fig. 9) nor for BCC metals. For simple materials (including liquid metals), the turnover of the growth velocity occurs at a relatively large undercooling of about $0.3T_m$[25] (0.19 – $0.25T_m$ in this study), compared to 0.05 – $0.1T_m$ for the complex materials[51,52], which are diffusion-controlled with a strong temperature dependence. According to my joint diffusion/collision model, only a fraction of the liquid atoms at the interface needs thermal activation for the growth of pure metals and so makes a smaller contribution to slow down the interface kinetics. Therefore, the turnover of growth velocity is simply caused by the interplay of increasing driving force and decreasing interface kinetics, as do the complex materials[26,51,52], but at a larger $\Delta T$. Meanwhile, the remaining liquid atoms solidify through the collision-limited mode without thermal activation barrier, even at high undercooling. This explains the large growth velocity of pure metals including FCC,

BCC and HCP metals at $T_c$[5,6], as well as the crystallization of pure FCC metal of Pb ($T_m = 600.6$ K) at a temperature as low as 4 K[7].

The current model provides a general guide to the research in the crystal growth of simple materials, for instance it reveals that both FCC and BCC metals have a similar growth kinetics. This helps us to unpuzzle the odd experimental observation of Zhong et al.[13]. It is partly attributed to high $T_m$ of the refractory BCC metals: 2183 K (V), 2896 K (Mo), 3290 K (Ta) and 3695 K (W) on one hand, and a wide range of $T_m$ of these FCC metals: 933 K (Al), 1235 K (Ag), 1337 K (Au), 1828 K (Pd), 2237 K (Rh) and 2739 K (Ir) on the other hand. With ultrafast liquid quenching technique, it is likely to achieve a transition from the joint diffusion/collision mode to the diffusion-controlled mode at large $\Delta T$ for the studied BCC metals (e.g., $T_{tran} = 1040$ K for Ta) due to an increase in $x_{therm}$ with increasing $\Delta T$, as well as for these FCC metals with high $T_m$ (e.g., Rh, Ir). The glass transition temperature, $T_g$, of Ta MG is about 1650 K[13], and so it is reasonable for $x_{therm} = 1$ below $T_{tran}$. Such a transition will not expect for either FCC (e.g., Al, Ag, Au) or other BCC metals with low $T_m$, as demonstrated in this study with Al, where $x_{therm} = 0.71$ at room temperature. Our prediction with this analytical model may have some discrepancy from the actual $x_{therm}$, but the clear margin of the calculated $x_{therm}$ at room temperature between the

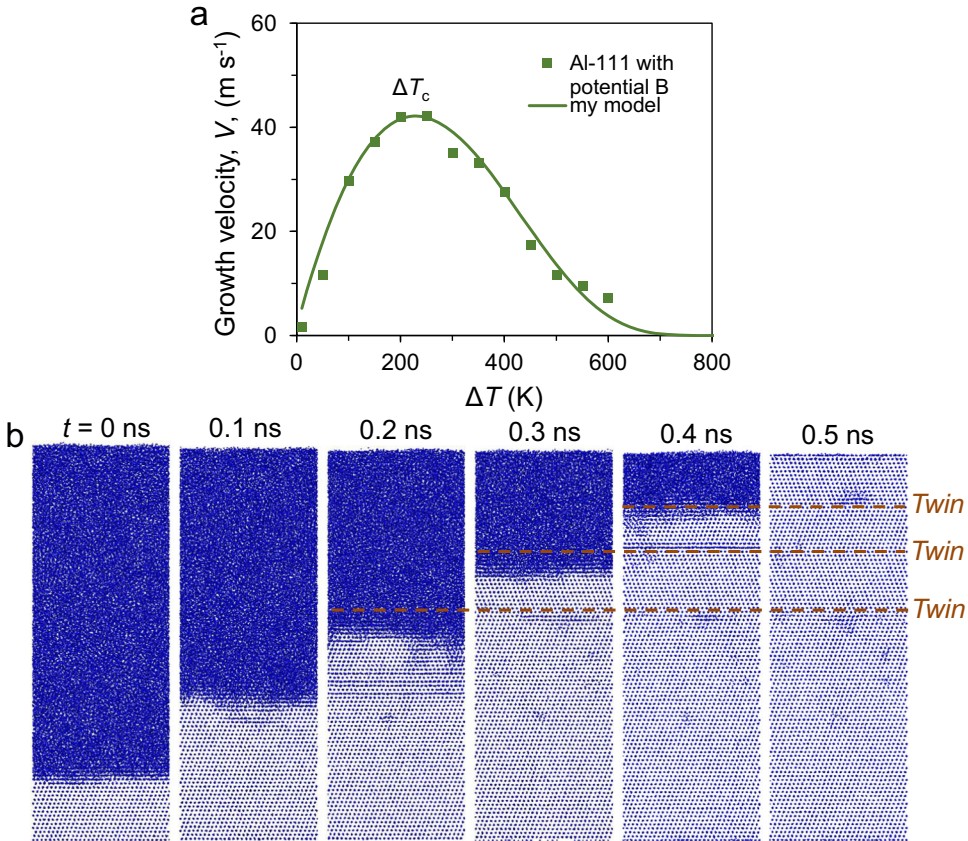

**Fig. 5 | Crystal growth at (111) interface. (a)** Growth velocity, *V*, as a function of the undercooling (Δ*T*) and (**b**) front views of the snapshots from the simulation time *t* = 0 to 0.5 ns for the Al (111) interface during the simulation at *T* = 637.7 K (Δ*T* = 300 K) with Potential B (embedded-atom method (EAM) potential for aluminium, developed by Song and Mendelev (see Methods)). The fitting with our model (see below) is also included in (**a**). The growth velocity reaches the maximum value at the crossover undercooling, Δ*T*$_c$ = 230 K. Stacking faults and twin boundaries form (dashed lines in (**b**)), where the stacking faults may die out later. (Source data are provided as a Source Data file).

metals with high and low $T_m$ is consistent with the experimental observations. Thus, the diffusion-controlled process at ambient temperature is not the ultimate reason for successful manufacture of the monatomic MGs from the BCC metals but not from FCC metals[13]. It should be pointed out for the vitrification of V a coating of an amorphous carbon layer 5 nm thick is needed to protect V from oxidation[13], where V has the lowest $T_m$ in these BCC metals, also lower than Rh and Ir of FCC metals. It is reasonable to suspect that the coating itself might help V to be vitrified. On the other hand, the solid clusters with close-packed FCC/HCP-like signature are dominant, especially for the sub-critical clusters, in the deeply undercooled liquid metals[53–59], as confirmed by the electronic structures that both liquid Ta, Mo, W, Nb of BCC metals and liquid Pd of FCC metal have the FCC-like short-range order[60]. The FCC-like clusters cannot be accommodated at the L/S interface of the BCC metals with open-packed structure, promoting formation of the MGs at extremely high cooling rate. However, they can easily attach to the FCC crystals at the interface, possibly with so-called glass-to-crystal (GC) mode[52]. The GC growth mode suggests that a diffusionless crystal growth proceeds several orders of magnitude faster than the extrapolated growth rates from the diffusion-controlled regime in the supercooled liquid for some simple organic glass formers[61]. It is attributed to the spatially heterogeneous dynamics[62], which seems to be in accordance with the FCC-like clusters in deeply undercooled liquid of FCC metals. This could be the reason that Zhong et al.[13] failed to produce the monoatomic MGs of FCC metals. Ref. 26 proposes an interface wetting effect that the compatible solid clusters in the growth front can promote the growth by wetting the liquid-crystal interface due to the reduction in the interfacial energy between

the cluster and solid phase, which may underlie the GC growth mechanism. This hypothesis can be evidenced by the phenomenon that some impurities (such as O and C) in undercooled liquid Ni enhances its glass forming ability possibly by changing the FCC-like clusters into the dissimilar one. For instance, it reported that nanometre-sized Ni MG was produced by a splash quenching technique with a cooling rate of $10^{10}$ K s$^{-1}$[63], and later turned out that the impurity had the responsibility otherwise it was not possible to vitrify Ni of FCC metal with this technique[64]. It is also noted that the monatomic W MG is not stable at ambient temperature although W of BCC metal has the highest $T_m$ in all the studied metals. Thermodynamics calculation reveals that the Gibbs free energy of FCC W becomes positive below about 550 K (Supplementary Fig. 10), and consequently the dominant FCC-like clusters at small Δ*T* will relax to the BCC-like clusters at high Δ*T*. The W MG undergoes spontaneous crystal growth at the glass/crystal interface (GCI) to a well-defined BCC structure at ambient temperature, possibly with the GC mode. Therefore, the GC mode may be applicable in some specific cases for the crystal growth of pure metals, which in turn provides a unique opportunity to establish the mechanism of GC growth due to the simplicity of the structures of metals as suggested by this study and ref. 26.

I have demonstrated that a considerable fraction of liquid atoms at the interface is thermal activated during the growth of pure liquid metals, and the joint diffusion/collision model is better to describe the growth kinetics by manifesting both collision and diffusion growth modes. My model is sufficiently robust to predict the general growth behaviour of pure metals, revealing FCC and BCC metals have a similar growth kinetics. It is able to quantitively describe the temperature

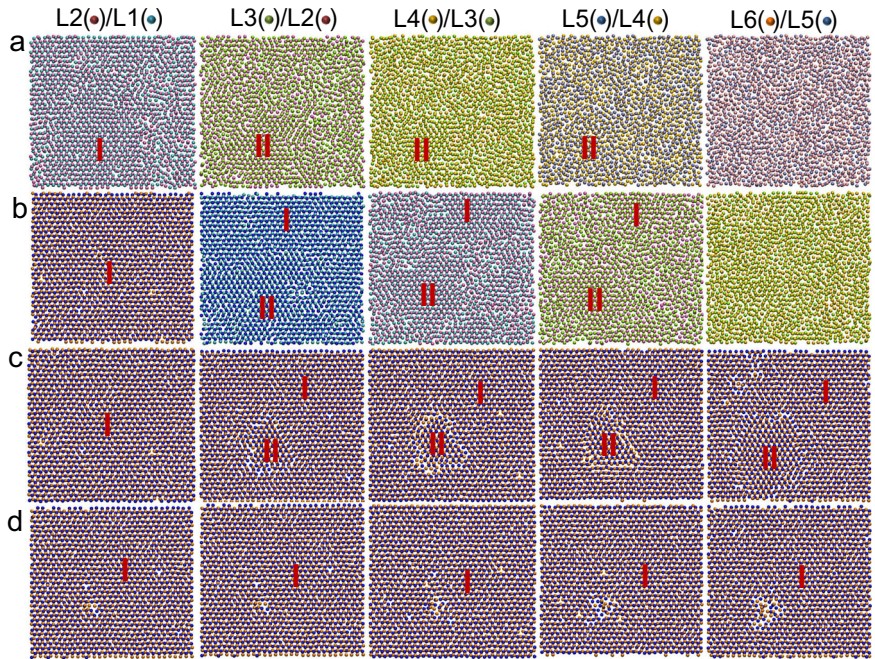

**Fig. 6 | Dislocation correction mechanism in crystal growth of (111) interface.** Top views of the snapshots of the interfacial layers in the Al (111) system (**a**) at the simulation time $t = 0.16$ ns, (**b**) 0.17 ns, (**c**) 0.2 ns, and (**d**) 0.22 ns during the simulation at $T = 687.7$ K (undercooling $\Delta T = 250$ K) with Potential B. L1 (cyan spheres), L2 (mauve spheres), L3 (yellow spheres), L4 (orange spheres), L5 (iceblue spheres) and L6 (pink spheres) denote the 1st to 6th interfacial liquid layers, respectively, at $t = 0.16$ ns, which gradually transform to solid layers with $t$. I and II represent the regions with normal and wrong templating sequences, respectively. The wrong templating appears at L3/L2 at 0.16 ns, leading to formation of the stacking fault, which died out at 0.22 ns through dislocation motions (referred to as dislocation correction mechanism).

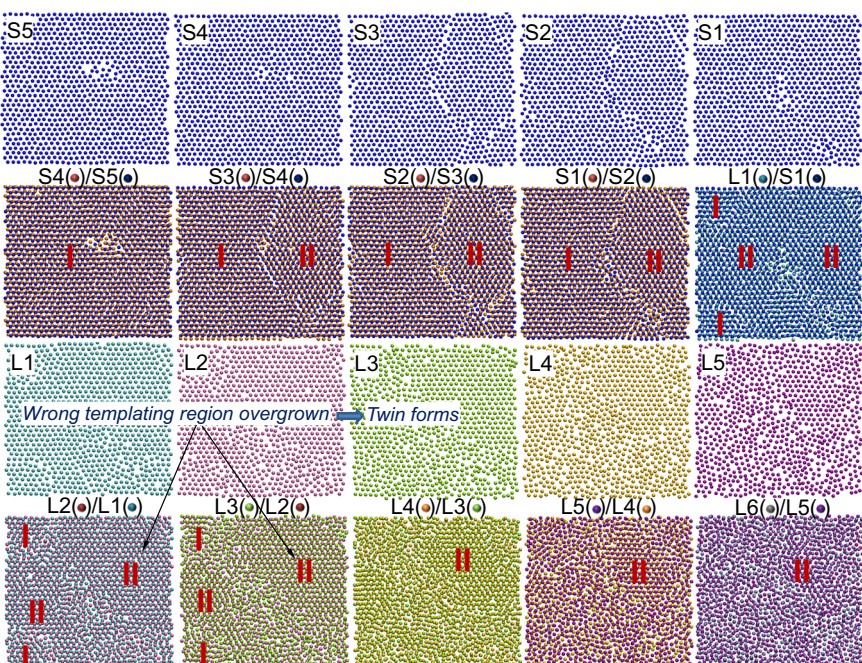

**Fig. 7 | Formation of twin boundary in crystal growth of (111) interface.** Top views of the snapshots from the 5th solid layer (S5) to 6th liquid layer (L6) at interface in the Al(111) system at the simulation time $t = 0.26$ ns during the simulation at $T = 637.7$ K (undercooling $\Delta T = 300$ K) with Potential B, where S1 (blue spheres), L1 (cyan spheres), L2 (mauve spheres), L3 (yellow spheres), L4 (orange spheres), L5 (purple spheres) and L6 (silver spheres) denote the 1st solid layer to 6th liquid layer, respectively, at the interface. The wrong templating (II) region overgrows the normal templating (I) region in the L1/S1, leading to formation of a twin boundary.

dependence of growth kinetics, as well as properly interpret the turnover of growth velocity, the fast growth at $T_c$, the crystallization of FCC metal of Pb at very low temperature and the odd experimental observations of vitrification of single-element metallic liquids. This study has significant implications not only in advancing our understanding of the solidification theory, but also in practice such as modelling of solidification, phase change materials and lithium dendrite growth in lithium-ion battery with an extension of this model.

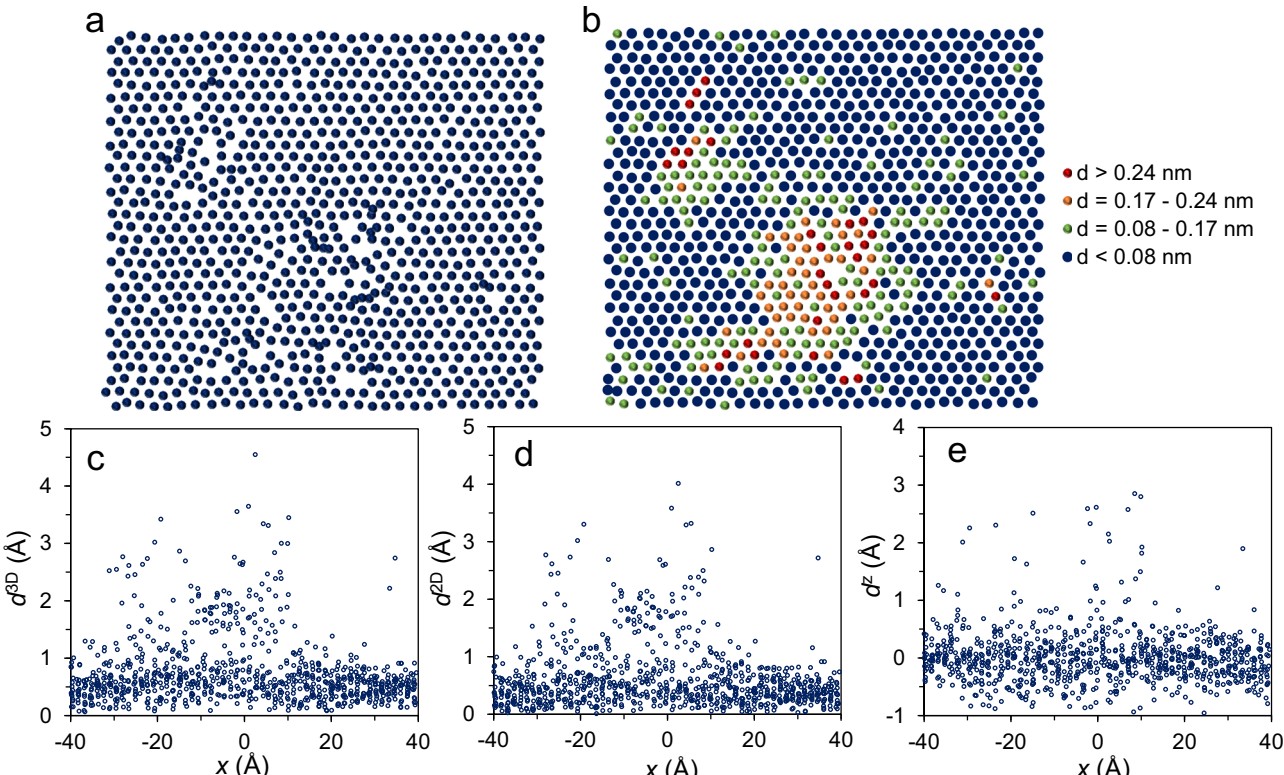

**Fig. 8 | Atomic displacements at (111) interface.** Top views of the snapshots of the 1st liquid interfacial layer (L1) in the Al (111) system (**a**) at the simulation time $t = 60$ and (**b**) 69 ps during the simulation at $T = 782$ K with Potential A, as well as atomic displacements in (**c**) 3-dimensional displacement ($d^{3D}$), (**d**) 2-dimensional displacement ($d^{2D}$), and (**e**) z-directional displacement ($d^z$), from $t = 0.06$ to 0.069 ns for these atoms located in the L1 at 0.069 ns. The atoms in (**b**) are coloured according to the $d^{3D}$. (Source data are provided as a Source Data file).

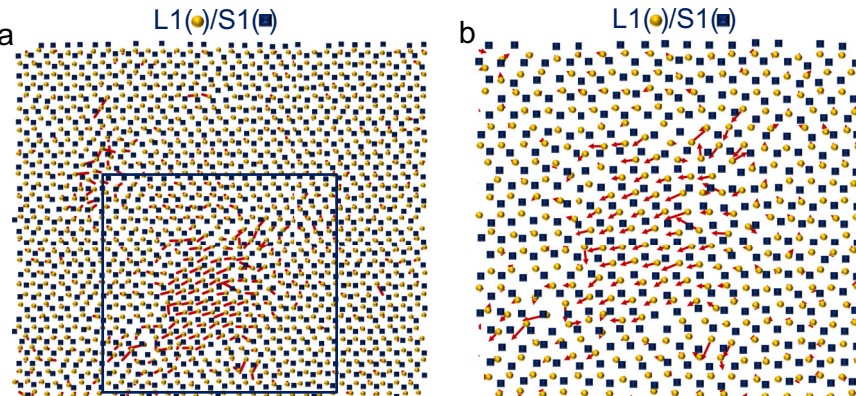

**Fig. 9 | Atomic displacements from C position to B position in crystal growth of (111) interface.** (**a**) Top view of the snapshot of the 1st liquid interfacial layer (L1 (orange spheres)) imposed on the 1st solid interfacial layer (S1 (blue squares)) and (**b**) its enlarged part in the Al (111) system at the simulation time $t = 0.06$ ns during the simulation at $T = 782$ K (undercooling $\Delta T = 160$ K) with Potential A. The S1 is fully solid at 0.06 ns. The arrows (or short lines) denote the displacement vectors of the atoms, located in the L1 at $t = 0.06$ ns, from $t = 0.06$ to 0.069 ns. The box in (**a**) is used to highlight the region that corresponds to (**b**). It noted that the significant fraction of atoms moved from C position at 0.06 ns to B position at 0.069 ns, with energy barriers to overcome.

## Methods

### Crystal growth simulation

MD simulations were performed to investigate the crystal growth of Al(111), (110) and (100) interfaces, with 46,080, 54,000 and 48,000 atoms, respectively, in the simulation systems, using DL_POLY_4.08 MD package[65]. The embedded-atom method (EAM) potential for aluminium, developed by Zope and Mishin[66] (denoted as Potential A), was used to model the interatomic interactions. To validate the simulation, the simulation on a system of Al(111) interface with double size in z-direction (92,160 atoms in the simulation system) was carried out using LAMMPS[67] with the EAM potential, developed by Song and Mendelev[68] (denoted as Potential B). Periodic boundary conditions are imposed in x- and y-directions, and a region of vacuum with an extent of 60 Å is inserted with periodic boundary conditions in z-direction. During the simulation, half of the simulation system was pinned and the rest was melted by heating it to 1500 K with the Nose-Hoover NVT ensemble. The system was equilibrated at a temperature near melting temperature, $T_m$, and then relaxed with the NVE ensemble by setting all the atoms free

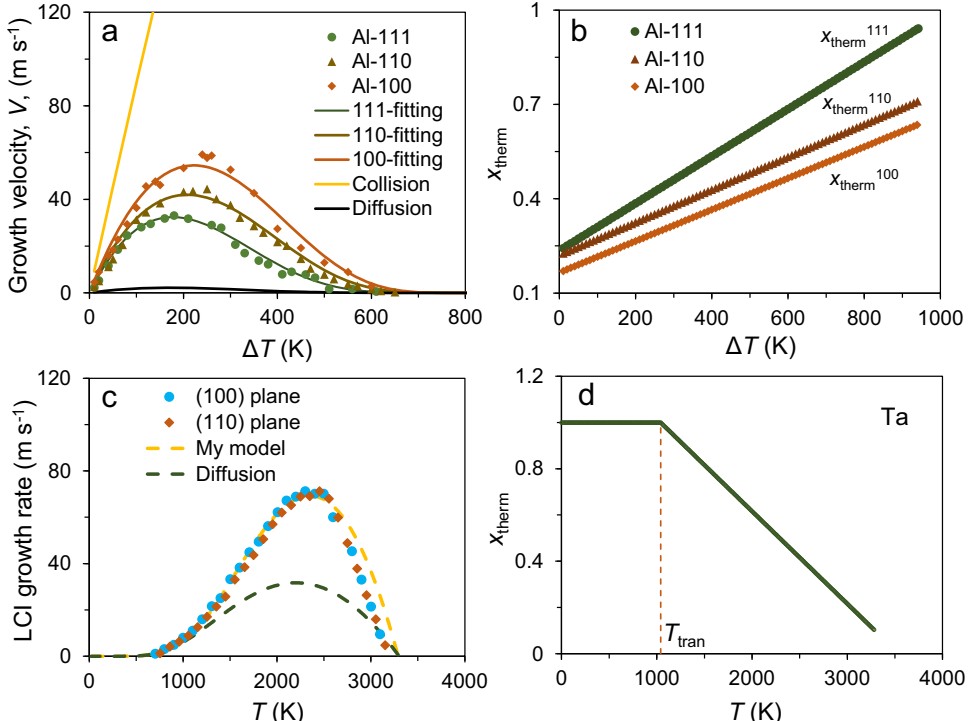

**Fig. 10 | Comparison of crystal growth models. (a)** The simulated growth velocity, $V$, is plotted as a function of the undercooling ($\Delta T$) for the Al (111), (110) and (100) systems with Potential A, as well as the fittings with collision, diffusion and my models, **(b)** the fraction, $x_{\text{therm}}$, of liquid atoms with thermal activation in crystal growth is plotted as a function of $\Delta T$, **(c)** crystal growth rate at the liquid-crystal interfaces (LCIs) of Ta (100) and (110) systems on the basis of classic simulation from ref. 13, as well as the plots with diffusion model and the fitting with my model, and **(d)** the fraction of liquid atoms with thermal activation, $x_{\text{therm}}$, obtained from the fitting with my model as a function of temperature $T$. $T_{\text{tran}}$ is the transition temperature (dashed line) from the joint diffusion/collision mode to diffusion mode. (Source data are provided as a Source Data file).

to find the accurate $T_{\text{m}}$. The simulations on crystal growth were then performed under varied undercooling from 10 K to 650 K with the Nose-Hoover NVT ensemble, using a time step of 0.001 ps.

### Characterization of atomic layering
The atomic layering at the L/S interface is characterized by the atomic density profile, $\rho(z)$[69]:

$$\rho(z) = \frac{<N_z>}{L_x L_y \Delta z},$$ (5)

where $z$ is the distance away from the center of the system, $N_z$ is the number of atoms in a bin between $z - \Delta z/2$ and $z + \Delta z/2$ at simulation time $t$, $\Delta z$ is the width of the bin, which is a 10th of the layer spacing in this study, and $L_x$ and $L_y$ are the $x$ and $y$ dimensions of the system. The angled brackets indicate a time-averaged quantity.

### Local bond-order analysis
The local bond-order analysis was performed to label solid and liquid atoms in the simulation system as an input to the training of machine learning. The local bond-order parameter is calculated as[31]:

$$\mathbf{q}_l(i) = \left( \frac{4\pi}{2l+1} \sum_{m=-l}^{l} |\mathbf{q}_{lm}(i)|^2 \right)^{\frac{1}{2}},$$ (6)

where the $(2l+1)$ dimensional complex vector $\mathbf{q}_{lm}(i)$ is a sum of the spherical harmonics, $Y_{lm}(r_{ij})$, over all the nearest neighbours $j$ of the atom $i$. The atoms $i$ and $j$ are recognised to be connected by a crystal-like bond if $\mathbf{q}_6(i) \cdot \mathbf{q}_6(j) > q_{\text{cut}}$, where $q_{\text{cut}} = 0.7$, an optimized hyperparameter in the training of machine learning. The fraction of the crystal-

like bonds between the atoms $i$ and $j$ is defined as the order parameter, $\alpha$, to label the atoms. The atoms $i$ is labelled as solid for $\alpha > \alpha_{\text{thres}}$ and liquid for $\alpha < \alpha_{\text{thres}}$, where $\alpha_{\text{thres}} = 0.4$, another optimized hyperparameter in the machine learning.

### Machine learning
Both the support vector machine (SVM) and neural networks (NN) classifiers were employed in the machine learning to identify solid and liquid atoms in the simulation systems. The local structure of each atom $i$ was characterized using a set of radial structure function, $G_i(r)$[38,39]:

$$G_i(r) = \sum_{j=1}^{n(i)} \exp\left( -\frac{(r_{ij} - r)^2}{2\sigma^2} \right),$$ (7)

where $n(i)$ is the number of the neighbours of atom $i$ within a cutoff radius $r_{\text{cut}}$, and $r_{ij}$ is the distance between atom $i$ and its neighbour $j$, and $r$ and $\sigma$ are the parameters to define the $G_i(r)$. With a set of 23 radial structure functions (23 is an optimal hyperparameter), the 23-dimensional space ($\mathbb{R}^{23}$) was built by the local-structure fingerprint, $\mathbf{x}_i$, for each atom $i$ in the simulation system:

$$\mathbf{x}_i = (G_i(r_0), G_i(r_1) \ldots G_i(r_{22})).$$ (8)

The atom $i$ in the simulation system receives label $y_i = 1$ for $\alpha >= 0.4$, and $y_i = -1$ for $\alpha < 0.4$. The data was assembled by pairing the label $y_i$ with its corresponding structural fingerprint $\mathbf{x}_i$. The $\mathbf{x}_i$, averaged 5 snapshots of the Al (111) system during the simulation at $T = 942$ K with Potential A, was used to find the hyperparameters in the machine learning with a grid-search algorithm[40]. It turned out $\sigma = 0.4$ Å, $r_{\text{cut}} = 3.35$ Å, $r_n = (2.0 + 1.2n)$ Å with $n = 0, 1, 2, \ldots, 22$, with an accuracy

of 93% and 95% above, respectively, for the SVM and NN classifiers. The optimal regularization parameter $C$ for penalty is 1.0 for the SVM, and the optimal hidden layers is 5 for the NN. With the optimized hyperparameters, the SVM and NN classifiers were used to train a data set of 0.5 million data point obtained from the steady state growth during the simulations of the Al(111), (110) and (100) interfaces in a temperature range between $T = 932$ K and 640 K, which was standardized to have zero mean and standard deviation of one. The solid and liquid atoms in the simulation system were then distinguished with the trained SVM and NN classifiers, respectively.

The softness, $S$, of each atom $i$ was calculated with the trained SVM. The $S$ is the signed distance from the hyperplane, $S_i = w^* \cdot \mathbf{x}_i - b^*$, where $w^*$ and $b^*$ are the parameters that define the hyperplane. It has been used to classify liquid and solid atoms in the crystal growth[27] and to characterize the local structure of the glass, revealing it is strongly correlated with the glass transition dynamics[70].

The principal component analysis (PCA)[40] was carried out to obtain a set of successive orthogonal components of each data point with a maximum amount of the variance. The $t$-distributed Stochastic Neighbour Embedding ($t$-SNE) method[41,42] was used to perform different transformations in different regions of $\mathbb{R}^{23}$ to find a balance between the local and global aspects of the $\mathbf{x}_i$ distribution. By trial and error, we choose the first 18 PCA components to pass to the $t$-SNE with a perplexity of 200, which gives a faithful representation of the data points in a two-dimensional space.

### Calculation of growth velocity

The steady-state growth velocity, $V$, is firstly calculated from the variation in the number of solid atoms, $N_S$, with time, $t$, as[8]:

$$V = \frac{dN_S/dt}{\rho_S A}, \tag{9}$$

where $\rho_S$ is the number density in the bulk solid, and $A$ is the lateral area of the simulation system. The $N_S$ is obtained by performing the machine learning with the trained NN. Also, the $V$ was measured from the derivative of the total enthalpy, $H$, of the simulation system versus $t$[25]:

$$V = \frac{a}{L}\frac{\partial H}{\partial t}, \tag{10}$$

where $a$ is the atomic layer spacing, and $L$ is the latent heat at corresponding temperature. The calculated $V$ with two methods agrees within a few percent.

## Data availability

The datasets generated in this study have been deposited in the Brunel University London database, Figshare [https://doi.org/10.17633/rd.brunel.26029045.v1][71]. Additional raw data can be found in Source Data file. Source data are provided with this paper.

## Code availability

The code used in the current study has been deposited in Code Ocean [https://doi.org/10.24433/CO.8127284.v1][72].

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

## Acknowledgements

The author would like to give thanks to Prof. Brian Cantor and Dr. Chamini Mendis for their constructive suggestions and reading this manuscript, and also to Dr. Feng Gao for providing thermodynamic data of Al and Ta. The EPSRC was gratefully acknowledged for providing financial support under Grant No. EP/N007638/1 and EP/S036296/1. We also were grateful to the UK Materials and Molecular Modelling Hub for computational resources, which is partially funded by EPSRC (Grant

Nos. EP/P020194/1 and EP/T022213/1), and maintained with support from Brunel University London.

## Author contributions

Hua Men made all the contributions.

## Competing interests

The author declares no competing interests.
