## [Peer Review File · Nature Communications]

A joint diffusion/collision model for crystal growth in pure liquid metalsReviewer #1 (Remarks to the Author):

It has been a long-standing question on the detailed kinetics of atomic attachment at the solid/melt interface during solidification. The manuscript presents a step forward in revealing the fundamental process behind solidification, by running molecular dynamics simulations and analyzing the solidification velocity in conjunction with pre-existing theoretical models. It is proposed that the liquid atoms at the interface can be classified into thermally activated and athermal ones, and an improved model is developed to describe the solidification velocity against undercooling. The findings help to advance the understanding of the solidification process and are inspiring to related scientific issues. It can be recommended for acceptance.

It is intriguing to ascribe the thermal activated nature to liquid diffusion and migration of non-in-registry atoms. While one fundamental issue remains unsolved: why should fcc have higher fraction of athermal than bcc, particularly in bcc there is no "in-registry" issue for the solid/melt interface.

Reviewer #2 (Remarks to the Author):

The article "Whether or not Crystal Growth Needs Thermal Activation in Pure Liquid Metals" covers topics related to the theory of solidification, focusing on crystal growth in pure liquid metals. This topic is interesting and the research methods used are up to date in Reviewer's opinion. However, Reviewer requests clarification of several issues and making corrections, which are listed below in order to increase the clarity of the manuscript.

Discussion remarks:

1. In Reviewer's opinion, the author should mention in the abstract that the study examined the mechanism of liquid atoms at Al (111), (110) and (100) 111 interfaces.
2. Reviewer suggests that the abstract should briefly describe the assumptions of the conducted experiments described in the methodology regarding the heating and cooling process, f.e. that in the simulation half of the system was pinned and the rest was heated to a temperature of 1500 K, then the system was equilibrated at a temperature close to the melting point, and crystal growth simulation was performed at different undercooling from 10 K to 650 K using a time step of 0.001 ps.
3. A significant part of the literature comes from the last century and the early 2000s - that is twenty and more years ago. Reviewer is aware that some of the items refer to physical formulas and theories. However, has there been no attempt in recent years to conduct similar research as described in the presented article? „Recently, a challenge to the collision theory has been posed by a study of Zhong et al. [13].“- publication by Zhong et al. it's from 2014, so it is already 10 years old, so it is nothing new.
4. "It is noted that the number of atoms per layer, NL, increases continuously across the interface from the liquid to the solid (Fig. 1a). This is also true for Al(110) and (100) interfaces (Fig. 1b and c) (...)". Is it possible to mark the liquid and solid areas in Fig. 1a-c? Interpreting the NL increase along liquid to solid in the figure would be simpler.
5. What are the causes of the differences in T_m in Fig. 1a-c?
6. In the abstract and at the end of the discussion it is written that the study is useful for modeling the growth of lithium dendrites in a lithium-ion battery. Reviewer asks for a more detailed development of this issue (f.e. how to model dendrite growth in this type of batteries) and to raise the issue of implementing these calculations in practice in the "Introduction" section.
7. In the opinion of Reviewer, the conclusions (if it is certainly the last paragraph before the „Methods“) are too general and incomplete, f.e. without direct reference to the results obtained:
- „ We have demonstrated that a considerable fraction of liquid atoms at the interface is thermal activated during the growth of pure liquid metals, and the joint collision/diffusion model is better to describe the growth kinetics by manifesting both collision and diffusion growth modes.“-why?
- „ Our model is robust to predict the general growth behaviour of pure metals, revealing FCC and BCC metals have a similar growth kinetics.“-why?

Editorial remarks:

1. In Reviewer's opinion, the division into the article and supplementary information together with

the use of different font colors creates chaos and makes the article difficult to read. Moreover, the article submitted for review would be easier to read if the illustrations were directly below the text.

2. Reviewer does not feel entitled to criticize the linguistic correctness, however, is it correct to write "we" in an article with one author?

3. After "Abstract" there is immediately "Results". The name of the chapter "Introduction" is missing - please enter it.

4. The "Conclusions" is also missing.

5. "Turnbull proposed (...) [3]" - please add the reference after the name as follows: "Turnbull [3] (...)". Please do the same elsewhere in the text.

Reviewer #3 (Remarks to the Author):

In this study, Men introduced a novel growth model to explain the growth rates of pure metal crystals from their liquids during undercooling. This model is particularly original because it integrates the well-established diffusion-controlled model by Wilson and Frenkel (WF) with the thermal activation-independent collision-limited model. The author employed classical Molecular Dynamics to examine the growth behavior of an FCC metal, specifically aluminum (Al), and utilized machine learning techniques to distinguish between barrier-free atomic attachment and the thermally activated growth mechanism caused by incorrect lattice templating. This approach yielded a quantitative model capable of describing the temperature dependence of growth kinetics, along with several intriguing phenomena related to crystal growth.

One limitation of the study is its focus primarily on FCC systems, which differ in atomic packing from other crystal types, such as BCC metals. While the study attempts to fit the growth rate of Ta (a BCC metal) using the proposed model, it remains ambiguous whether the apparent accuracy of this fit stems from the model's inherent flexibility—owing to additional fitting parameters—or if it genuinely reflects the underlying atomic growth mechanisms. Thus, it is important for the author to further substantiate their model by providing concrete evidence supporting both the "collision-limited" and "diffusion-controlled" growth modes across different crystal types. Such validation is essential, given the prevailing belief that growth kinetics and lattice growth activation can significantly vary between BCC and FCC systems, especially in terms of thermally activated growth and the implications of incorrect lattice templating.

The author sought to elucidate the mechanisms underlying the formation of monoatomic metallic glasses in body-centered cubic (BCC) systems, as reported by Zhong et al., in the context of the combined collision/diffusion model. However, the explanations presented raise several flags.

- The reference to "T = 1040 K" on page 7, line 339, is unclear. It is essential to specify what the temperature refers to in this context.

- The manuscript suggests that the formation of metallic glass can be partly ascribed to the high melting temperatures of BCC metals, arguing that the parameter x_{therm} increases with undercooling. This reasoning is challenging to follow, as x_{therm} is influenced by other variables in their model as well. For low melting-point metals, x_{therm} could be large at small undercoolings.

- On page 8, line 347, the assertion that "the solid clusters with close-packed face-centered cubic (FCC)/hexagonal close-packed (HCP)-like signatures are predominant in undercooled liquid metals" is questioned. Based on current understanding, atomic arrangement in undercooled pure metallic clusters typically does not exhibit a majority in FCC or HCP-like packing, except a minor fraction of subcritical nuclei, especially in the case of BCC metal liquids. The sources cited (references 48 and 49) do not substantiate this claim. Additionally, the interpretation of electronic structural alterations suggested in reference 50, indicating a shift from a BCC-like to an FCC-like short-range order, necessitates further verification.

- The hypothesis of a transition from glass to crystal (GC) growth mode to explain the formation of FCC crystals from the liquid phase is not convincing. Such a discussion not only deviates from the

core findings of the manuscript but also appears speculative. To convincingly argue for this growth mode, direct evidence should be provided, either from simulations or experimental work.

Despite these deficiencies, this work enhances our understanding of pure metal growth and introduces new insights into the growth mechanisms of FCC metals. If the author can adequately address the concerns regarding BCC system studies mentioned above, I would support the publication of this paper in Nature Communications, given its substantial contribution to the field of crystal growth.

Reviewer #4 (Remarks to the Author):

In this paper, the author presents the results of a detailed simulation study of the crystal growth from molten aluminium in which they argue that the imperfect correlation in the interface between the crystal structure and the adjacent liquid results in some fraction of the attachment kinetics being barrierless while the remaining fraction requires activation. A modified expression from the crystal growth rate is presented in which this fraction of activated growth appears explicitly as a parameter to be fitted.

This is a well written paper that explores an interesting and clearly articulated picture of the crystal growth from the melt. That said, there are a number of significant issues with this paper, described below, such that I cannot recommend publication in its current form.

1. I have a problem with the new expression from the growth rate V in Eq.4 (and its derivation in Part 7 of the Supplementary Material). The author argues that crystal growth is governed by two processes: a fast barrierless one involving the attachment of particles that are correctly registered to the lattice and a slow activated one involving those particles in the stacking fault positions. Generally, when a kinetic process consists of two processes, the slower one is rate determining and so dominates. This is not what is expressed in Eq. 4. The derivation of this equation assumes that the accumulation of particles in the wrong position has no influence on the rate of deposition of particles in the right position but this doesn't seem physically correct. The only way a particle can be designated as being in the 'correct' position is with reference to the bit of the interface close by. If that is disordered due to the inclusion of particles with the wrong registry then this will modify the local definition of what registry is 'correct'. This effect is missing in Eq.4.

2. The empirical evidence for the importance of activation rests heavily on the low temperature data i.e. at and below the turnover in V . My difficulty with this is that the nucleation rate in liquids of pure FCC forming liquids at these large supercoolings is typically extremely high. Crystal growth is only meaningful if the supercooled melt into which the crystal is growing remains a well defined metastable state. How is it that nucleation is avoided in these calculations? The presence of crystal nuclei misaligned with the growing crystal will give rise to a slow down of the crystal front via a mechanism quite distinct from that associated with growth into the metastable liquid. To attach significance to this slow down it is important to explicitly rule out the possible impact of nucleation occurring in advance of the growing front.

Responses to Reviewers

Reviewer #1

Question 1: It is intriguing to ascribe the thermal activated nature to liquid diffusion and migration of non-in-registry atoms. While one fundamental issue remains unsolved: why should fcc have higher fraction of athermal than bcc, particularly in bcc there is no "in-registry" issue for the solid/melt interface.

Response: The question might be “why should fcc have higher fraction of thermal activated atoms than bcc at the solid/melt interface?”. FCC metals have the close-packed structure and BCC has the open structure. For the FCC metals, the (111) orientation is densely packed and therefore has a higher fraction of thermal activated atoms due to the "in-registry" issue. On the other hand, the (110) and (100) orientations are less densely packed, without the "in-registry" issue, and so have a smaller fraction of thermal activated atoms. Our study reveals that the thermal activated fraction is 0.24, 0.23, 0.17, respectively, for the Al(111), (110) and (100) orientations at an undercooling of 10 K. For the BCC metals, the (110) and (100) orientations are less densely packed, similar to the counterpart of FCC metals, and therefore have a smaller fraction of thermal activated atoms. For instance, the thermal activated fraction is 0.1 for the (110) and (100) orientations of Ta at an undercooling of 10 K.

I have made further explanation to clarify the issue in Discussion section of the revised manuscript.

Reviewer #2

Question 1: In Reviewer's opinion, the author should mention in the abstract that the study examined the mechanism of liquid atoms at Al (111), (110) and (100) 111 interfaces.

Response: I agree with the Reviewer #2, and have added “interfaces of Al (111), (110) and (100)” in the abstract in the revised manuscript.

Question 2: Reviewer suggests that the abstract should briefly describe the assumptions of the conducted experiments described in the methodology regarding the heating and cooling process, f.e. that in the simulation half of the system was pinned and the rest was heated to a temperature of 1500 K, then the system was equilibrated at a temperature close to the melting point, and crystal growth simulation was performed at different undercooling from 10 K to 650 K using a time step of 0.001 ps.

Response: I appreciate the suggestion of Reviewer #2 to give a more detailed description of the simulation approach used in this study in the Abstract section. However, the journal suggests an abstract of approximately 150 words. The usage of every word in the Abstract has been carefully considered in the case of this manuscript that the words of Abstract already exceeds the suggestion of the journal.

Question 3. A significant part of the literature comes from the last century and the early 2000s - that is twenty and more years ago. Reviewer is aware that some of the items refer to physical formulas and theories. However, has there been no attempt in recent years to conduct similar research as described in the presented article? “Recently, a challenge to the collision theory has been posed by a study of Zhong et al. [13].” - publication by Zhong et al. it's from 2014, so it is already 10 years old, so it is nothing new.

Response: I agree with Reviewer #2's point that “2014” literally is not “recently”, and has deleted “recently” in the revised manuscript. But indeed, there is intensive disputes about the kinetics of crystal growth in the **recent** years, as I have pointed out in the literature review [4,8-21,23-27]. Here, I just name a few. In the publication of “The mechanism of the ultrafast

crystal growth of pure metals from their melts. *Nat. Mater.* 17, 881 (2018)”, Sun et al. [18] suggests to have explicitly resolved the origin of the barrierless growth kinetics by using computer simulations of crystallization in pure metals, in favour of collision theory. R. Freitas and E. J. Reed [27] proposed the local-structure dependent (LSD) crystal growth model, in favour of diffusion-controlled mechanism (Freitas, R. & Reed, E. J. Uncovering the effects of interface-induced ordering of liquid on crystal growth using machine learning. *Nat. Commun.* 11, 3260 (2020)). Hu, Y. -C. & Tanaka, H. [26] report that suppressing the crystal-like structural order in the supercooled liquid through a new order-killing strategy can reduce the crystallisation rate by several orders of magnitude, in contrary to the study of R. Freitas and E. J. Reed (Hu, Y. -C. & Tanaka, H. Revealing the role of liquid preordering in crystallisation of supercooled liquids. *Nat. Commun.* 13, 4519 (2022)). All these researches were well performed, and the simulation results were rigorous. But their interpretation on the simulation results are usually based on either collision or diffusion model, and thus the mechanism of crystal growth still confuses the research community so far. I hope that the researches on the growth kinetics in this research community can move forward from the present study.

Question 4: “It is noted that the number of atoms per layer, N_L , increases continuously across the interface from the liquid to the solid (Fig. 1a). This is also true for Al(110) and (100) interfaces (Fig. 1b and c) (...)”. Is it possible to mark the liquid and solid areas in Fig. 1a-c? Interpreting the N_L increase along liquid to solid in the figure would be simpler.

Response: I appreciate the suggestion of Reviewer #2, and have marked the liquid and solid areas in Fig. 1(a)-(c) for better interpreting the N_L increase across the L/S interface in the revised manuscript.

Question 5: What are the causes of the differences in T_m in Fig. 1a-c?

Response: The melting point of metals has a strong dependence on the anisotropy of crystallographic orientation, due to the variation in the density of packing of atoms in a particular crystal plane. The surface atoms on the planes with the least atomic packing density have the maximal amplitude of atomic vibrations. According to Lindemann’s criterion [R1], the melting point, T_m , of a metal is directly proportional to the Debye temperature, θ_D : [R2]

$$\theta_D^2 = C \langle u^2 \rangle^{-1} [T_m/M]$$

where $\langle u^2 \rangle$ is mean-square amplitude of atomic vibration, M is atomic weight, and C , a constant, is equal to $9h^2/k_B$ with h denoting the reduced Planck’s constant and k_B the Boltzmann’s constant. Debye temperature is closely related to the amplitude of vibration, the lower with the less closely packed planes. For FCC structure, the (111) interface is densely packed and the (110) and (100) interfaces are less densely packed. Therefore, the (111) orientation of FCC metals has a higher T_m than (110) and (100) orientations.

I have given a further explanation about the causes of the differences in T_m in Fig. 1(a)-(c) in Rough liquid/solid interfaces, Results section of the revised manuscript.

References:

[R1] Lindemann, F. A. About the calculation of molecular own frequencies, *Phys. Mag.* 11, 609 (1910).

[R2] Chatterjee, B. Anisotropy of melting for cubic metals, *Nature* 275, 203 (1978)

Question 6: In the abstract and at the end of the discussion it is written that the study is useful for modeling the growth of lithium dendrites in a lithium-ion battery. Reviewer asks for

a more detailed development of this issue (f.e. how to model dendrite growth in this type of batteries) and to raise the issue of implementing these calculations in practice in the "Introduction" section.

Response: Lithium metal is considered one of the most promising anode materials for application in next-generation Li-ion batteries [R3,R4]. However, some issues, such as: uncontrolled dendrite formation, large volume changes and irreversible electrolyte degradation reactions inherent in lithium-metal-based batteries, can result in severe safety concerns and low Coulombic efficiency. The Li dendrite growth occurs at the electrolyte/anode interface in the repeated lithium deposition and dissolution processes during charge/discharge cycling, where the fundamental interfacial mechanism of dendrite growth is not yet fully understood. Lithium dendrite formation is the combined consequence of the microstructure and chemical interactions in the solid electrolyte interphase (SEI) domains, and an anode roughening process may be correlated with the beginning stage of dendrite formation [R4]. In the present study, I found that the planar growth in pure liquid metal at small undercooling changes to the multi-spherical growth under high undercooling, which causes abnormally fast growth and produces apparent slope discontinuity of the growth velocity. The roughening of the growing front under high undercooling observed in the present study may be not in exactly the same situation as the lithium dendrite growth, but could share the same mechanism. Therefore, our finding would be helpful for the modelling of the growth of lithium dendrites in the Li-ion battery.

I have made further explanation in the Discussion section, rather than the Introduction section, in the revised manuscript since this part is an extension of our present study but doesn't directly contribute to clarification of the growth kinetics.

References:

[R3] L. A. Selis and J. M. Seminario, Dendrite formation in Li-metal anodes: an atomistic molecular dynamics study, *RSC Adv.* 9, 27835 (2019).

[R4] H. G. Lee, S. Y. Kim and J. S. Lee, Dynamic observation of dendrite growth on lithium metal anode during battery charging/discharging cycles, *npj Comput. Mater.* 8, 103 (2022).

Question 7: *In the opinion of Reviewer, the conclusions (if it is certainly the last paragraph before the "Methods") are too general and incomplete, f.e. without direct reference to the results obtained:- "We have demonstrated that a considerable fraction of liquid atoms at the interface is thermal activated during the growth of pure liquid metals, and the joint collision/diffusion model is better to describe the growth kinetics by manifesting both collision and diffusion growth modes."-why? - "Our model is robust to predict the general growth behaviour of pure metals, revealing FCC and BCC metals have a similar growth kinetics."-why?*

Response: In the present study, the joint collision/diffusion model is developed based on the reasonable physical model and logical mathematical analysis, as presented more detailed in the revised manuscript. With this model, I analysed the MD simulation data in this study and literature [13], and drew the solid conclusion that in general both collision and diffusion mechanisms are involved in the crystal growth of pure metals. Thus, the present study resolved the long-standing fundamental question in understanding the interface kinetics of crystal growth: whether or not crystal growth needs thermal activation in pure liquid metals. Our joint collision/diffusion model can reasonably describe the growth kinetics of both FCC and BCC metals, and provide a proper interpretation about the temperature dependence of growth kinetics, turnover of growth velocity, extremely large growth velocity, crystallization of pure FCC metal at very low temperature and vitrification of single-element metallic

liquids). Therefore, our collision/diffusion model is better to describe the growth kinetics, and robust to predict the general growth behaviour of simple metals.

Our conclusion (Summary section in the revised manuscript) is directly referenced in the Results and Discussion sections. In the section of **Results: Thermal activation in crystal growth**, I state “a considerable fraction of liquid atoms at the (111) interface needs thermal activation during the growth. across the Al(111) interface at $\Delta T = 160$ K, about one third of liquid atoms is thermal activated in the growth, behaving in the diffusion-controlled mode”. This is the direct reference to the **Conclusion** of “We have demonstrated that a considerable fraction of liquid atoms at the interface is thermal activated during the growth of pure liquid metals”.

In the section of **Results: Predicting growth velocity**, I state “With a single fitting parameter of x_{therm} , our joint collision/diffusion model can reasonably predict the growth velocity for the Al (111), (110) and (100) interfaces, as shown in **Figs. 5a and 10a**” and “Our model also can fit the simulated growth velocity of BCC metals, such as Ta, with the data recovered from the study of Zhong *et al.* [13], as shown in **Fig. 10c.**” In the section of **Discussion**, I state “Our model provides a general guide to the research in the crystal growth of simple materials, for instance it reveals that both FCC and BCC metals have a similar growth kinetics.” This is the direct reference to the **Conclusion** of “the joint collision/diffusion model is better to describe the growth kinetics by manifesting both collision and diffusion growth modes.” and “Our model is robust to predict the general growth behaviour of pure metals, revealing FCC and BCC metals have a similar growth kinetics.”

Question 8: Editorial remarks:

(1). *In Reviewer’s opinion, the division into the article and supplementary information together with the use of different font colors creates chaos and makes the article difficult to read. Moreover, the article submitted for review would be easier to read if the illustrations were directly below the text.*

Response: The identical font “Times New Roman” with black colour and size of 12 is used in both article and Supplementary information of this manuscript. I have embedded the illustrations in the text in the revised manuscript for the convenience of reviewing process.

(2). *Reviewer does not feel entitled to criticize the linguistic correctness, however, is it correct to write “we” in an article with one author?*

Response: I have changed “we” to “I” in the revised manuscript.

(3). *After “Abstract” there is immediately “Results”. The name of the chapter “Introduction” is missing - please enter it.*

Response: I followed the general format of the publications for Nature Communications, and the headlines of “Introduction” and “Conclusion” have been included in the revised manuscript.

(4). *The “Conclusions” is also missing.*

Response: It is the same as (3).

(5). *“Turnbull proposed (...) [3]” - please add the reference after the name as follows: “Turbull [3] (...)”. Please do the same elsewhere in the text.*

Response: I agree with the Reviewer #2, and have added the reference after the name of the author in the revised manuscript.

Reviewer #3

Question 1: *One limitation of the study is its focus primarily on FCC systems, which differ in atomic packing from other crystal types, such as BCC metals. While the study attempts to fit the growth rate of Ta (a BCC metal) using the proposed model, it remains ambiguous whether the apparent accuracy of this fit stems from the model's inherent flexibility—owing to additional fitting parameters—or if it genuinely reflects the underlying atomic growth mechanisms. Thus, it is important for the author to further substantiate their model by providing concrete evidence supporting both the "collision-limited" and "diffusion-controlled" growth modes across different crystal types. Such validation is essential, given the prevailing belief that growth kinetics and lattice growth activation can significantly vary between BCC and FCC systems, especially in terms of thermally activated growth and the implications of incorrect lattice templating.*

Response: In developing the joint collision/diffusion model, it doesn't need any assumption for a specific crystal structure. Therefore, our model should be applicable for all the simple metals, and it genuinely reflects the underlying atomic growth mechanism. In this study, I choose Al as the model system, and the reason is that Al as FCC metal has the close-packed structure. The Al(111) is the densely packed plane, and the Al(110) and (100) are the less densely packed planes. Thus, the anisotropy in the growth kinetics of Al is only attributed to the structural factor, and the chemical effect can be excluded. Simultaneously, the crystal growth of Al has been extensively investigated in literature. Therefore, the simulations on FCC Al might provide the best data to testify our model. Our model is then used to fit the data of MD simulation on Ta of BCC metals and it produces reasonable agreement. Indeed, BCC metals have the open structure, different from FCC metals. All the low-index planes (such as: (100) and (110)) of BCC metals are less densely packed, similar to the FCC(110) and (100) plane at the microstructural scale. During the crystal growth, the densely packed FCC(111) plane is most stable in all the low-index planes, and on the other hand the low-index planes of BCC metals are always less densely packed. This might be largely attributed to the difference in the growth kinetics between FCC and BCC metals.

In fitting the MD simulation data with our joint collision/diffusion model, the only fitting parameter is x_{therm} , the atomic fraction of the interfacial atoms that needs thermal activation in the growth. Our model doesn't adopt any more flexibility in the fitting, even less than these works in the literature, but produces considerably good accuracy. The reason is that both FCC and BCC metals, as well as all the simple materials, generally exhibit very similar growth behaviours and our model genuinely reflects the underlying atomic growth mechanism. In this sense, our joint collision/diffusion model has been well validated and it is not necessary to repeat these simulations work on FCC and BCC metals having been performed many times in literature.

I have given further explanation to reflect the point of Reviewer #3 in the revised manuscript.

Question 2: *The author sought to elucidate the mechanisms underlying the formation of monoatomic metallic glasses in body-centered cubic (BCC) systems, as reported by Zhong et al., in the context of the combined collision/diffusion model. However, the explanations presented raise several flags.*

(1) - *The reference to "T = 1040 K" on page 7, line 339, is unclear. It is essential to specify what the temperature refers to in this context.*

Response: “ $T = 1040$ K” refers to the transition temperature from the joint to diffusion mode in the crystal growth of Ta with increasing the undercooling. I have specified the transition temperature as T_{tran} in the revised manuscript.

(2)- *The manuscript suggests that the formation of metallic glass can be partly ascribed to the high melting temperatures of BCC metals, arguing that the parameter x_{therm} increases with undercooling. This reasoning is challenging to follow, as x_{therm} is influenced by other variables in their model as well. For low melting-point metals, x_{therm} could be large at small undercoolings.*

Response: For the metals with low T_m , such as Pb, the crystallization can occur at a temperature as low as 4 K. This is one of the experimental evidences for the collision-limited model [7], suggesting that the growth for a considerable fraction of atoms is athermal, i.e., $x_{\text{therm}} < 1$, at the growing front of either FCC or BCC metals at very low temperature. On the other hand, the nanometer-sized monatomic metallic glasses (MGs) of BCC metals (Ta, V, Mo) have been successfully produced, stable at ambient temperature, with a high liquid-quench technique to achieve an ultrahigh cooling rate of 10^{14} K/s [13]. These refractory BCC metals have a high T_m . The glass transition temperature, T_g , of Ta MG is about 1650 K, indicating that $x_{\text{therm}} = 1$ at a temperature below T_g but well above room temperature. These studies suggest that the x_{therm} could differ significantly at an identical temperature, e.g., room temperature, for these metals with low and high T_m , and so the prediction with our model is generally consistent with these experimental observations. I agree on part of the point of the Reviewer #3 that “ x_{therm} could be large at small undercoolings for low melting point metals”. Our prediction with this analytical model may have some discrepancy from the actual x_{therm} (it is understandable), but it genuinely reflects the underlying atomic growth mechanism.

It should be pointed out that the experimental result of Zhong et al. [13] can not be interpreted properly by any other models. They successfully manufactured the MGs for the BCC metals with high T_m , which has been attributed to the diffusion-controlled mechanism, but failed for the FCC metals due to the collision-limited growth mode. It appears that BCC and FCC metals should take different growth mechanisms. However, one expects that Zhong et al. can not produce the MGs for the BCC metals with low T_m with the same techniques. The question is what growth mechanism these BCC metals with low T_m should take. Actually, both FCC and BCC metals exhibit very similar growth behaviour, as do other simple materials, and our joint collision/diffusion model has provided reasonable explanation for the experimental result of Zhong et al. [13] and other experimental phenomena.

I have given further explanation to reflect the point of Reviewer #3 in Discussion section of the revised manuscript.

(3)- *On page 8, line 347, the assertion that "the solid clusters with close-packed face-centered cubic (FCC)/hexagonal close-packed (HCP)-like signatures are predominant in undercooled liquid metals" is questioned. Based on current understanding, atomic arrangement in undercooled pure metallic clusters typically does not exhibit a majority in FCC or HCP-like packing, except a minor fraction of subcritical nuclei, especially in the case of BCC metal liquids. The sources cited (references 48 and 49) do not substantiate this claim. Additionally, the interpretation of electronic structural alterations suggested in reference 50, indicating a shift from a BCC-like to an FCC-like short-range order, necessitates further verification.*

Response: For the liquids of simple metals at small and mild undercooling, I agree with the Reviewer #3’s point “atomic arrangement in undercooled pure metallic clusters typically

does not exhibit a majority in FCC or HCP-like packing, except a minor fraction of subcritical nuclei". In this case, the size of solid clusters in the undercooled liquid is small and the fraction of the atoms in the solid clusters is not significant. Actually, the determination of the structure for these solid clusters with small size is a bit ambiguous, and it often leads to confusion in literature. The majority of the atoms in the undercooled liquid can be classified as individual atoms with disordered structure. The attachment of these individual liquid atom to the solid surface is dominant in the growth kinetics. With increasing the undercooling, however, the fraction of atoms in the solid clusters becomes more and more significant, where both the size and quantity of solid clusters increase. The attachment of the atoms in these solid clusters will involve the dissolution of the solid clusters, and so the rearrangement of local structure is needed for the growth to proceed, i.e., the growth is diffusion-controlled. This might be the main reason that the x_{therm} increases with decreasing the undercooling. At high undercooling, the attachment of these solid clusters with subcritical size to the solid surface expects to become dominant, for which the size of subcritical nuclei can reach a few nanometres and the determination of the structure is more reliable. This is the reason that I refer to Refs. 48 and 49 for clarifying the structure of solid clusters. While the structure of the subcritical clusters is compatible with solid phase, it will promote the growth kinetics, through the interface wetting effect as proposed by Ref. 26, in the case of FCC metals. The interface wetting effect [26] may underlie the mechanism of GC model [47]. As a consequent, the FCC metals, even with high T_m , can not be vitrified. If it is not compatible, whatever it is ico-cluster (as indicated by Ref.26) or FCC/HCP clusters (as suggested in this study) in Ta, in the case of BCC metals, the growth front of solid phase can be blocked by the solid clusters during the growth at high undercooling. The rearrangement of local structure is needed for the growth to proceed through dissolution of the solid clusters, i.e., the growth is diffusion-controlled. This might be unlikely at high undercooling for the BCC metals with high T_m , which is so can be vitrified. Ref. 50 indicates that the structure of solid cluster in the undercooled liquid of BCC metals might be different from the bulk solid, as shown in Ref. 26, and we admit that it is very difficult to determine the exact structure of solid cluster, particularly for these with small size at small undercooling.

I have given further explanation to reflect the point of Reviewer #3 in Discussion section of the revised manuscript.

(4)- The hypothesis of a transition from glass to crystal (GC) growth mode to explain the formation of FCC crystals from the liquid phase is not convincing. Such a discussion not only deviates from the core findings of the manuscript but also appears speculative. To convincingly argue for this growth mode, direct evidence should be provided, either from simulations or experimental work.

Response: I agree part of the point of Reviewer #3 that I should provide convincingly argument for the GC growth mode in this manuscript. As I pointed out in the response to **Question 3**, Ref. 26 provides an interface wetting mechanism which suggests that the compatible solid clusters in the growth front can attach to the solid surface as a whole to promote the growth due to the reduction in the interfacial energy between the cluster and solid phase. This could happen for the FCC metals with high T_m at high undercooling, where the attachment of solid clusters is overwhelming at the growing front due to the large fraction of atoms in the solid clusters in the undercooled liquid. Thereby, the interface wetting mechanism might underlie the mechanism of the GC growth mode, which suggests a "diffusionless" crystal growth proceeds several orders of magnitude faster than the extrapolated growth rates from the diffusion-controlled regime in the supercooled liquid for some simple organic glass formers [51]. The GC crystal growth is attribute to the spatially

nano-scaled heterogeneous dynamics [52], which seems to be in accordance with the FCC-like clusters in deeply undercooled liquid of FCC metals. Thus, the research in the crystal growth of simple metals [26 and this study] provides the convincing evidence to elucidate the mechanism of the GC growth model, which remains puzzled in the research community of organic glass formation.

It should be pointed out that Ref. 26 proposes a reasonable mechanism for fast growth of the simple metals, but the understanding of the interface wetting effect on growth kinetics may need to be further considered, where a new factor, $P(\gamma(T))$, is incorporated to WF model to measure the effect of the liquid structural ordering on the growth kinetic. The present study indicates that the x_{therm} , the fraction of thermal activated atoms at the growing front, should be responsible for the growth kinetics. On the other hand, the authors in Ref.26 suggest that “lots of ico-like orderings in Ta” and “ico-like ordering should hinder the growth of crystal-like preorders into the liquid region. This feature explains the slower crystal growth rate and better glass-forming ability of Ta than Zr”. This is agreed with our present study that the solid clusters in the supercooled liquid is not compatible with the BCC structure, whatever ico-like or FCC-like, and so the BCC metals with high T_m can be vitrified.

I have given further explanation to reflect the point of Reviewer #3 in Discussion section of the revised manuscript.

Reviewer #4

Question 1: *I have a problem with the new expression from the growth rate V in Eq.4 (and its derivation in Part 7 of the Supplementary Material). The author argues that crystal growth is governed by two processes: a fast barrierless one involving the attachment of particles that are correctly registered to the lattice and a slow activated one involving those particles in the stacking fault positions. Generally, when a kinetic process consists of two processes, the slower one is rate determining and so dominates. This is not what is expressed in Eq. 4. The derivation of this equation assumes that the accumulation of particles in the wrong position has no influence on the rate of deposition of particles in the right position but this doesn't seem physically correct. The only way a particle can be designated as being in the 'correct' position is with reference to the bit of the interface close by. If that is disordered due to the inclusion of particles with the wrong registry then this will modify the local definition of what registry is 'correct'. This effect is missing in Eq.4.*

Response: In developing the joint collision/diffusion model, I have considered the contribution to the growth kinetics from all the N atoms both with correct and wrong registry. The m atoms with wrong registry have an activation energy $Q > 0$ eV and these $(N-m+1)$ atoms with correct registry have an of $Q_c = 0$ eV. Thus, the probability of attachment for all N atoms is $\exp(-Q/k_B T)^{m/N} \cdot \exp(-Q_c/k_B T)^{(N-m+1)/N} = \exp(-Q/k_B T)^{m/N}$, identical to that of the m atoms with wrong registry. This is the reason that only $\exp(-Q/k_B T)^{m/N}$ appears in my analytical model. I have given a more detailed description for the development of the joint collision/diffusion model in the revised manuscript, but it doesn't change the formulation of Eq. 4.

I have given more detailed description in developing the joint collision/diffusion model in Part 7 of the Supplementary Information of the revised manuscript.

Question 2: *The empirical evidence for the importance of activation rests heavily on the low temperature data i.e. at and below the turnover in V . My difficulty with this is that the nucleation rate in liquids of pure FCC forming liquids at these large supercoolings is*

typically extremely high. Crystal growth is only meaningful if the supercooled melt into which the crystal is growing remains a well defined metastable state. How is it that nucleation is avoided in these calculations? The presence of crystal nuclei misaligned with the growing crystal will give rise to a slow down of the crystal front via a mechanism quite distinct from that associated with growth into the metastable liquid. To attach significance to this slow down it is important to explicitly rule out the possible impact of nucleation occurring in advance of the growing front.

Response: I agree with the point of the Reviewer #4 about “The presence of crystal nuclei misaligned with the growing crystal will give rise to a slow down of the crystal front via a mechanism quite distinct from that associated with growth into the metastable liquid”. This phenomenon can not be ruled out for the crystal growth of pure metals at high undercooling. In my recent publication of “H. Men, A molecular dynamics study on the boundary between homogeneous and heterogeneous nucleation, *J. Chem. Phys.* 160, 094702 (2024)”, it reveals that homogeneous nucleation in pure liquid aluminium can occur at an undercooling of 457 K, which falls in the high undercooling range (maximum undercooling of 650 K) used in the present study. In this case, it is similar to the CET that the planar growth of the solid phase is blocked by the new grains from homogeneous nucleation due to the extremely high undercooling and transforms to spherical growth. However, this phenomenon has never been observed in our studies, or reported in literature. For instance, we observed that FCC Al continues to grow even at an undercooling of 600 K in all three orientations. Zhong et al reported that the BCC Ta was vitrified at a T_g of about 1650 K, but no crystalline phase was identified in the MG by either HRTEM observation or MD simulations. The reason might be that the crystal growth of pure metals is always easier than homogeneous nucleation in pure liquid metals since the crystalline phase can be considered as a perfect nucleating substrate. Homogeneous nucleation could occur only after some incubation time during the undercooling with the slow cooling rate (in my simulation liquid aluminium is cooled down in a time step of 5 ns). Otherwise, homogeneous nucleation can't be observed if liquid aluminium is cooled down in a larger cooling rate, such as a time step of 2 ns, and liquid aluminium is transformed to amorphous phase at 0 K. Therefore, homogeneous nucleation may not be observed in the undercooled liquid in the front of growing solid phase under current simulation or experimental conditions, where the size of simulation systems or the samples is usually limited to nano scale and so the extremely large undercooling or cooling rate is applied.

I have given further explanation to reflect the point of Reviewer #4 in Discussion section of the revised manuscript.

Reviewer #1 (Remarks to the Author):

The author has revised the manuscript according to the comments and suggestions of the reviewers, and the response seems to address the concerns reasonably well. I would like to recommend its acceptance for publication now.

Reviewer #2 (Remarks to the Author):

The Author answered the Reviewer's questions and made corrections to the manuscript. The Reviewer recommends the article "Whether or not Crystal Growth Needs Thermal Activation in Pure Liquid Metals" for publication.

Reviewer #3 (Remarks to the Author):

In the revised manuscript, the author has addressed some of the concerns raised by the referees, leading to an improvement in the paper's overall quality. However, several issues still warrant further clarification.

Firstly, the author asserts that the growth velocity data fitting relies solely on the x_{therm} parameter, as highlighted in Line 561. This claim appears to be inaccurate, given that x_{therm} 's dependency on undercooling is represented as a linear equation ($b \Delta T + c$), necessitating at least two parameters for an accurate fit of the growth velocity data. Please clarify or correct this.

Regarding the characterization of "solid atoms" in undercooled liquids, the term seems to be overstated. Typically, undercooled liquids are characterized by a lack of crystalline order among most atoms, although dynamical heterogeneity and local structural motifs, such as icosahedral ordering, are observed. The author's definition of atom softness, based on a machine-learning approach, introduces a hidden parameter, S (softness), whose physical interpretation remains ambiguous. While the S parameter of atoms in undercooled liquids spans a wide range ($-1 < S < 1$), overlapping with that of crystalline atoms, this does not imply a similarity in crystallographic features. This is analogous to observations made through Steinhardt bond-order-parameter analysis. Consequently, the designation of these atoms as "solid atoms" within the liquid phase lacks sufficient justification. Any references for that? Furthermore, the claim regarding the prevalence of solid clusters with FCC/HCP-like structures among solid atoms (in undercooled liquids) requires proper citation for verification.

The manuscript also presents an interesting rationale for the impossibility of forming monatomic metallic glasses (MGS) in low-melting-point metals. As mentioned in my previous report, X_{therm} is not directly related to the melting point of metal (and consequently undercooling) of the metal, but it is given by $b \Delta T + C$. The argument that only metals with a high melting point can sustain significant undercooling (ΔT) to achieve a notable x_{therm} value overlooks the potential impact of the coefficient b . If b is sufficiently large, a low-melting-point metal could theoretically achieve an x_{therm} value close to 1, challenging the presented logic, unless the author demonstrates that the b parameter is similar for all metals. More clarification on this would be beneficial.

Lastly, the manuscript would benefit significantly from a comprehensive grammar and spelling review. For example, in the abstract, notable errors include:

- "therm activated" should be corrected to "thermally activated."
- The phrase "need the thermal activation for growth" should be revised to "need thermal activation for growth."
- The expression "attach to the crystal without energy barrier" should be adjusted to "without an energy barrier."
- "Quantatively" should be corrected to "quantitatively."

Reviewer #4 (Remarks to the Author):

The authors has addressed the issues raised in the review. I recommend that the paper be accepted for publication without further change.

Response to Reviewer #3

Question 1: Firstly, the author asserts that the growth velocity data fitting relies solely on the x_{therm} parameter, as highlighted in Line 561. This claim appears to be inaccurate, given that x_{therm} 's dependency on undercooling is represented as a linear equation ($b \Delta T + c$), necessitating at least two parameters for an accurate fit of the growth velocity data. Please clarify or correct this.

Response: The fitting parameter $x_{\text{therm}} = b\Delta T + c$ is used in our collision/diffusion model. I have shown the physical origin of the linear relationship between x_{therm} and ΔT in Response 2 and 3. It should be pointed out that our joint collision/diffusion model can be deduced from either the collision-limited (Jackson) model or diffusion-controlled (WF) model, based on the reasonable physical model. Therefore, it genuinely reflects the underlying atomic growth mechanism.

Change: I considered the point of the Reviewer #3 and have deleted the “single” in the revised manuscript.

Question 2: Regarding the characterization of "solid atoms" in undercooled liquids, the term seems to be overstated. Typically, undercooled liquids are characterized by a lack of crystalline order among most atoms, although dynamical heterogeneity and local structural motifs, such as icosahedral ordering, are observed. The author's definition of atom softness, based on a machine-learning approach, introduces a hidden parameter, S (softness), whose physical interpretation remains ambiguous. While the S parameter of atoms in undercooled liquids spans a wide range ($-1 < S < 1$), overlapping with that of crystalline atoms, this does not imply a similarity in crystallographic features. This is analogous to observations made through Steinhardt bond-order-parameter analysis. Consequently, the designation of these atoms as "solid atoms" within the liquid phase lacks sufficient justification. Any references for that? Furthermore, the claim regarding the prevalence of solid clusters with FCC/HCP-like structures among solid atoms (in undercooled liquids) requires proper citation for verification.

Response: The softness, S , is defined as the signed distance from the hyperplane, $S_i = w^* \cdot x_i - b^*$, where w^* and b^* are the parameters that defines the hyperplane, based on the local structure of each atom i in this study. This parameter is calculated with SVMs (Support Vector Machines) of the machine learning, and used for classification. This approach has been demonstrated to be very robust in the varied practices of machine learning. In the context of this research, the parameter S itself has no physical meaning, and instead indicates the confidence scores of the classification for the individual atom as liquid or solid atom in the simulation system. It works pretty well for both bulk liquid and solid, with a little bit more uncertainty for the liquid/solid interface (The Steinhardt bond-order-parameter analysis has the same problem, which employs a threshold with some uncertainty to determine the solid/liquid connection between neighbouring atoms). In this study, we can achieve an accuracy of 93% in the prediction with the SVMs for the liquid/solid systems equilibrated at the melting point, which is reasonably good. In this case, nearly all the atoms have $S > 0$ in the bulk solid and $S < 0$ in the bulk liquid. The parameter S has been used to classify the liquid and solid atoms in the crystal growth [Ref. 27], and to characterize the local structure of the glass, revealing it is strongly correlated with dynamics of glass transition [R1]. Therefore, the designation of these atoms as "solid atoms" within the liquid phase in this study has sufficient justification.

There exists some controversy over the types (FCC, HCP, BCC, Icosahedral) of solid clusters in the undercooled liquid in literature. It found that the clusters with BCC-like structure

should be favoured in all simple fluids at small undercooling [R2,R3] and the formation of FCC nuclei is preferred at large undercooling ($> 0.5T_m$) [R4-R8, Refs.58&59]. At small undercooling, the solid clusters in the undercooled liquid are not only small in size but also scarce in quantity. For small clusters of the new phase, most of the atoms reside in the interfacial region [R9]. Interestingly, Wolde et al., reported that the surface atoms of the clusters may have the BCC structure while the core has an FCC structure. Thus, there exists large uncertainty in determining the structure of small clusters. Further, some studies suggest that while most of the small clusters may have the BCC or icosahedral structure (at small undercooling) the FCC/HCP structure will become dominant in the large clusters (at high undercooling) [R6]. My present study suggests that the solid cluster is barely observable in the bulk liquid Al at the melting point by either the local bond-order analysis (Supplementary Fig. 8) or the prediction with machine learning (as indicated by the softness S in Fig. 3a). At small undercooling, the number of the atoms in the solid clusters is negligible and its effect on the growth kinetics is negligible as well, no matter that these clusters have the structure of FCC, BCC, HCP or icosahedral. Both the number and size of solid clusters increase with increasing the undercooling. This is the reason that homogeneous nucleation should and must occur at high undercooling (at least have demonstrated by the atomistic simulations), where the critical size of the nuclei can reach the largest solid clusters in the undercooled liquid. Therefore, the number of atoms in the solid clusters increases with an increase in the undercooling (as indicated by the softness S in Fig. 3b). These atoms would make more and more significant contribution to slow down the growth kinetics at the interface since the rearrangement of local structures will need for these atoms to attach to the solid surface. As the literature [R4-R8, Refs.58&59] suggested, the FCC or HCP-like structure is dominant for the solid clusters at high undercooling. In the case that the structure of solid clusters is not compatible with the solidified phase (e.g., FCC/HCP clusters (or icosahedral inclusively) in BCC metals), these solid clusters have to resolve before they can attach to the solid. While the structure of solid clusters is compatible with the solidified phase, these solid clusters may need to readjust its orientation before the attachment as a whole or resolve (e.g., FCC/HCP clusters (or icosahedral inclusively) in FCC metals). Thus, the fraction of the thermally-activated atoms, x_{therm} , at the interface should increase with increasing the undercooling. This is the physical origin of the dependence of x_{therm} on the undercooling in the joint collision/diffusion model. While the x_{therm} is close to 1, the growth of BCC metals will be blocked by the solid clusters with subcritical size and dissimilar structure, and the undercooled liquid is transformed into glassy. In the case of FCC metals, on the other hand, the FCC-like clusters will facilitate the growth by the interface wetting mechanism as proposed by Ref. 26 (i.e., so-called GC mode in metals), due to the close contact between clusters, and so it is not successful to produce the metallic glass [Ref.13]. Thus, our joint collision/diffusion model provides a reasonable interpretation for these experimental results. **Change:** Following the suggestions of the Reviewer #3, and I have included the further reference to application of the softness S in the solidification research field and to the prevalence of solid clusters with FCC/HCP-like structures among solid atoms in undercooled liquids in the revised manuscript.

Question 3: The manuscript also presents an interesting rationale for the impossibility of forming monatomic metallic glasses (MGS) in low-melting-point metals. As mentioned in my previous report, x_{therm} is not directly related to the melting point of metal (and consequently undercooling) of the metal, but it is given by $b \cdot \Delta T + C$. The argument that only metals with a high melting point can sustain significant undercooling (ΔT) to achieve a notable x_{therm} value overlooks the potential impact of the coefficient b . If b is sufficiently large, a low-melting-point metal could theoretically achieve an x_{therm} value

close to 1, challenging the presented logic, unless the author demonstrates that the b parameter is similar for all metals. More clarification on this would be beneficial.

Response: From the theory of thermodynamic fluctuations, the probability of obtaining a cluster of the new phase containing n atoms, P_n , depends on the minimum reversible work for cluster formation, ΔG_n :

$$P_n \propto \exp(-\Delta G_n/k_B T) \quad (1)$$

where k_B is Boltzmann's constant. The equilibrium cluster size distribution per mole, N_n^e , is readily obtained from Eq. (1):

$$N_n^e = N_A \exp(-\Delta G_n/k_B T), \quad (2)$$

where N_A is Avogadro's number of molecules. Following the capillarity approximation, ΔG_n can be expressed as the sum of volume and interfacial energy contributions:

$$\Delta G_n = -n\Delta G' + (36\pi)^{1/3} v^{2/3} n^{2/3} \sigma, \quad (3)$$

where $\Delta G'$ is the Gibbs free energy per molecule of the new phase less than that of the initial phase, v is the molecular volume, and σ is the interfacial energy per unit area. At small undercooling (ΔT), $\Delta G'$ can be approximated as:

$$\Delta G' = \frac{\Delta H_f \Delta T}{N_A T_m}, \quad (4)$$

where ΔH_f is the heat of fusion and T_m is melting point. Then we have:

$$\begin{aligned} N_n^e &= N_A \exp\left(-\frac{-n\frac{\Delta H_f \Delta T}{N_A T_m} + (36\pi)^{1/3} v^{2/3} n^{2/3} \sigma}{k_B T}\right) = N_A \exp\left(-\frac{-n\frac{\Delta H_f \Delta T}{N_A T_m} + (36\pi)^{1/3} v^{2/3} n^{2/3} \sigma}{k_B (T_m - \Delta T)}\right) \\ &= N_A \exp\left(-\frac{-n\frac{\Delta H_f \Delta T}{N_A T_m} + (36\pi)^{1/3} v^{2/3} n^{2/3} \sigma}{k_B T_m (1 - \frac{\Delta T}{T_m})}\right) \\ &= N_A \exp\left(-\left(-n\frac{\Delta H_f \Delta T}{N_A T_m} + (36\pi)^{1/3} v^{2/3} n^{2/3} \sigma\right) \frac{1}{k_B T_m (1 - \frac{\Delta T}{T_m})}\right) \\ &= N_A \exp\left(-\left(-n\frac{\Delta H_f \Delta T}{N_A T_m} + (36\pi)^{1/3} v^{2/3} n^{2/3} \sigma\right) \frac{1}{k_B T_m} \left(1 + \frac{\Delta T}{T_m}\right)\right) \\ &= N_A \exp\left(-\frac{1}{T_m k_B} \left(-n\frac{\Delta H_f \Delta T}{N_A T_m} - n\frac{\Delta H_f \Delta T}{N_A T_m} \left(\frac{\Delta T}{T_m}\right)^2 + (36\pi)^{1/3} v^{2/3} n^{2/3} \sigma\right.\right. \\ &\quad \left.\left.+ (36\pi)^{1/3} v^{2/3} n^{2/3} \sigma \frac{\Delta T}{T_m}\right)\right) \\ &= N_A \exp\left(-\frac{(36\pi)^{1/3} v^{2/3} n^{2/3} \sigma}{k_B T_m} + \frac{1}{T_m k_B} \left(n\frac{\Delta H_f}{N_A} - (36\pi)^{1/3} v^{2/3} n^{2/3} \sigma\right) \frac{\Delta T}{T_m}\right) \\ &= N_A \exp\left(-\frac{(36\pi)^{1/3} v^{2/3} n^{2/3} \sigma}{k_B T_m}\right) \exp\left(\frac{1}{T_m k_B} \left(n\frac{\Delta H_f}{N_A} - (36\pi)^{1/3} v^{2/3} n^{2/3} \sigma\right) \frac{\Delta T}{T_m}\right) \\ &= N_A \exp\left(-\frac{(36\pi)^{1/3} v^{2/3} n^{2/3} \sigma}{k_B T_m}\right) \exp\left(\frac{\Delta T}{T_m} \frac{n\frac{\Delta H_f}{N_A} - (36\pi)^{1/3} v^{2/3} n^{2/3} \sigma}{T_m k_B}\right) \end{aligned}$$

$$\begin{aligned}
&= N_A \exp\left(-\frac{(36\pi)^{\frac{1}{3}}v^{\frac{1}{3}}n^{\frac{1}{3}}\sigma}{k_B T_m}\right) \left(1 + \frac{\Delta T}{T_m}\right)^{\frac{n \frac{\Delta H_f}{N_A} - (36\pi)^{\frac{1}{3}}v^{\frac{1}{3}}n^{\frac{1}{3}}\sigma}{T_m k_B}}. \\
&= N_A \exp\left(-\frac{(36\pi)^{\frac{1}{3}}v^{\frac{1}{3}}n^{\frac{1}{3}}\sigma}{k_B T_m}\right) T_m^{\frac{-n \frac{\Delta H_f}{N_A} + (36\pi)^{\frac{1}{3}}v^{\frac{1}{3}}n^{\frac{1}{3}}\sigma}{T_m k_B}} (T_m + \Delta T)^{\frac{n \frac{\Delta H_f}{N_A} - (36\pi)^{\frac{1}{3}}v^{\frac{1}{3}}n^{\frac{1}{3}}\sigma}{T_m k_B}}
\end{aligned} \quad (5)$$

Here, $\Delta T/T_m \approx 0$ at small undercooling, and so the following approximations can be applied:

$$\frac{1}{1 - \frac{\Delta T}{T_m}} = 1 + \frac{\Delta T}{T_m}, \quad (6)$$

$$\left(\frac{\Delta T}{T_m}\right)^2 = 0, \quad (7)$$

$$\exp\left(\frac{\Delta T}{T_m}\right) = 1 + \frac{\Delta T}{T_m}. \quad (8)$$

Also, it is noted that $n \frac{\Delta H_f}{N_A} - (36\pi)^{\frac{1}{3}}v^{\frac{1}{3}}n^{\frac{1}{3}}\sigma > 0$ holds true since the volume term of the free energy $n\Delta G' = n \frac{\Delta H_f \Delta T}{N_A T_m}$ should be close to the surface energy term $(36\pi)^{1/3}v^{2/3}n^{2/3}\sigma$ for the solid clusters and then a very small $\Delta T/T_m \approx 0$ is take away from the volume term $n\Delta G'$.

Eq. (5) indicates the equilibrium cluster size distribution, N_n^e , increases with increasing the undercooling ΔT , where the linear relationship between the N_n^e and ΔT is modulated by the terms of volume free energy and surface energy. It should be pointed out that Eq. (5) only holds true for small undercooling, but undoubtedly the N_n^e , increases with increasing ΔT . **Fig.1** shows the increase in the N_n^e with decreasing T for the lithium-disilicate glass as an example. This is consistent to our simulation result and the fitting result with our joint collision/diffusion model. It is according to the **statistic mechanics** that the atoms in the solid clusters in undercooled liquid increases with an increase in ΔT . This fact is well accepted in the research community of condensed physics, and I will not add the above deduction of Eq. (5) to the present manuscript. As mentioned in Response 2, these atoms in the solid clusters will make more and more significant contribution to the growth kinetics in the crystal growth with increasing the undercooling. Therefore, the fitting parameter of $x_{\text{therm}} = b \Delta T + c$ in our joint collision/diffusion model indeed reflects the genuine mechanism of crystal growth for pure simple materials. From our fitting, the b is 0.00075, 0.00052 and 0.0005, respectively, for Al(111), (110) and (100) interfaces (FCC metal) and 0.0004 for Ta(110) and (100) interfaces (BCC metal). It suggests that the b is very close for the metals, especially for the less densely packed interfaces of either FCC or BCC metals.

Fig.1 The steady-state cluster populations at 390 °C and 550 °C, calculated

using thermodynamic and kinetic parameters that are appropriate for crystallization of lithium-disilicate glass (Chapter 8). The critical sizes at the two temperatures are indicated. [R9]

Change: Following the suggestion of the Reviewer #3, and I have added the values of b from the fitting to the crystal growth of Al(111), (110), (100) interfaces and Ta(110, (100) interfaces to the revised manuscript.

Question 4: Lastly, the manuscript would benefit significantly from a comprehensive grammar and spelling review. For example, in the abstract, notable errors include:

- "therm activated" should be corrected to "thermally activated."
- The phrase "need the thermal activation for growth" should be revised to "need thermal activation for growth."
- The expression "attach to the crystal without energy barrier" should be adjusted to "without an energy barrier."
- "Quantatively" should be corrected to "quantitatively."

Response and Change: Following the comment and suggestions of the Reviewer #3, I have thoroughly conducted the grammar and spelling check, and also invited a native English expert, Dr. Chamini Mendis, to read this manuscript and made further improvement.

References:

- R1. Schoenholz, S., Cubuk, E., Sussman, D. et al. A structural approach to relaxation in glassy liquids. *Nature Phys.* **12**, 469–471 (2016). <https://doi.org/10.1038/nphys3644>.
- R2. Alexander S. & McTague, J. P. Should all crystals be bcc? Landau theory of solidification and crystal nucleation. *Phys. Rev. Lett.* **41**, 702 (1978).
- R3. ten Wolde, P. R. & Frenkel, D. Homogeneous nucleation and the Ostwald step rule. *Phys. Chem. Chem. Phys.* **1**, 2191-2196 (1999).
- R4. Mountain, R. D., & Brown, A. C. Molecular dynamics investigation of homogeneous nucleation for inverse power potential liquids and for a modified Lennard-Jones liquid. *J. Chem. Phys.* **80**, 2730 (1984).
- R5. Nosé, S., & Yonezawa, F. Isothermal–isobaric computer simulations of melting and crystallization of a Lennard-Jones system. *J. Chem. Phys.* **84**, 1803–1814 (1986).
- R6. Liu, J., Zhao, J. Z., & Hu, Z. Q. Kinetic details of the nucleation in supercooled liquid metals. *Appl. Phys. Lett.* **89**, 031903 (2006).
- R7. E, J. C., Wang, L., Cai, Y., Wu, H. A. & Luo, S. N. Crystallization in supercooled liquid Cu: Homogeneous nucleation and growth. *J. Chem. Phys.* **142**, 064704 (2015).
- R8. Zhou, L. L. Liu, R. S. Tian, Z. A. Liu, H. R. Hou, Z. Y. & Peng, P. Crystallization characteristics in supercooled liquid zinc during isothermal relaxation: A molecular dynamics simulation study. *Sci. Rep.* **6**, 31653 (2016). <https://doi.org/10.1038/srep31653>.
- R9. Kelton, K. F. & Greer, A. L. *Nucleation in Condensed Mater: Applications in Materials and Biology* (Oxford, UK: Pergamon, 2010).

Reviewer #3 (Remarks to the Author):

The referee has several concerns regarding the author's replies, particularly focusing on the justification for using "solid atoms" and the prevalence of HCP/FCC "solid clusters" in liquid phases. The citations provided by the author to support the use of "solid atoms" in undercooled liquids (Ref. 27 and R2) do not mention "solid atoms" at all. While the referee has no problem with characterizing atoms using a softness parameter, just that the term "solid atoms" raises some red flags.

In the reply, there also seems to be confusion between the structure of subnuclei and the structure of random clusters with a "negative" S softness. The interaction between the number of "solid clusters" and undercooling ΔT appears problematic. Analysis of the equation preceding Eq(5) suggests that N_e is determined by the ratio of ΔT to T_m , rather than undercooling alone. This challenges the author's claim that metallic glass formation is limited to metals capable of sustaining significant undercooling.

Despite these concerns, in principle, the referee does not object to the publication of the paper in NC.

Question (1): The referee has several concerns regarding the author's replies, particularly focusing on the justification for using "solid atoms" and the prevalence of HCP/FCC "solid clusters" in liquid phases. The citations provided by the author to support the use of "solid atoms" in undercooled liquids (Ref. 27 and R2) do not mention "solid atoms" at all. While the referee has no problem with characterizing atoms using a softness parameter, just that the term "solid atoms" raises some red flags.

Response: In Ref. 27 (R. Freitas, et al., *Nature Communications*, 11, 3260 (2020)), the liquid and **solid atoms** are labelled according to their softness (S) value (Fig. 4c), "where S is seen to capture the structural signs of dynamical heterogeneity in the **supercooled liquid** far from the crystal, with clear indications of strong spatial correlations. These fluctuating heterogeneities have recently been shown to be preferential sites for crystal nucleation (Paragraph 2, Page 2)." In [R2] (Alexander S., et al., *Phys. Rev. Lett.* 41, 702 (1978)) of our previous response, it states "at high temperature, and that, even where other structures are more stable, the first phase nucleated on rapid cooling can be bcc." We refer this paper to indicate that the BCC solid clusters could exist in the undercooled liquid at small undercooling, leading to the nucleation of BCC phase. Obviously, the existence of solid atoms (more precisely the atoms in solid clusters) in the undercooled liquid have been clearly indicated in both the papers.

Question (2): In the reply, there also seems to be confusion between the structure of subnuclei and the structure of random clusters with a "negative" S softness. The interaction between the number of "solid clusters" and undercooling ΔT appears problematic. Analysis of the equation preceding Eq(5) suggests that N_e is determined by the ratio of ΔT to T_m , rather than undercooling alone. This challenges the author's claim that metallic glass formation is limited to metals capable of sustaining significant undercooling.

Response: The structure of clusters is not random, and only these atoms with the crystal-like bonds are labelled as the solid cluster. So all these solid clusters have crystalline structure (though not perfect and/or mixed), but with a size less than the stable bulk phase. This is the reason that these atoms are labelled as solid atoms, which is distinct with liquid atoms with the disorder structure. The final formula of Eq. (5) has its physical meaning that the N_e will increase with increasing the undercooling while $T < T_m$ ($\Delta T > 0$), and decrease with increasing the overheating while $T > T_m$ ($\Delta T < 0$). The Reviewer pointed out that " N_e is determined by the ratio of ΔT to T_m , rather than undercooling alone." Actually, Eq. 5, suggests that for a specific metal the N_e is determined solely by the undercooling ΔT . In general, the N_e is determined by many more factors, such as: ΔT , T_m , molecular volume (v), interfacial energy per unit area (σ) and heat of fusion (ΔH_f). So the Reviewer's point can't be justified. Further, as we have pointed out in the previous response, Eq. (5) is only applicable for very small undercooling, but not for high undercooling.

Additionally, the crystallization can occur in Pb (with a low T_m of 600.6 K) at a temperature as low as 4 K [7], suggesting that some Pb atoms can attach to the solid by the collision mode without energy barrier (i.e., $x_{\text{therm}} < 1$). It indicates that some Pb atoms are still in a liquid status at a temperature close to 0 K. On the other hand, Ta and other metals with high T_m can be vitrified at a temperature of about 1500 K [13], suggesting that all the atoms must move by the diffusion mode with energy barrier (i.e., $x_{\text{therm}} = 1$). It indicates that all atoms are in a solid status at a temperature of 1500 K. We can conclude that the glass forming ability is relevant to the melting point, partially, as demonstrated in my current study, and my joint collision/diffusion model is consistent with the experimental observations.